# From Black Boxes to Transparent Minds: Evaluating and Enhancing the Theory of Mind in Multimodal Large Language Models

**Xinyang Li** [* 1]  **Siqi Liu** [* 1]  **Bochao Zou** [† 1]  **Jiansheng Chen** [1]  **Huimin Ma** [† 1]

## Abstract

As large language models evolve, there is growing anticipation that they will emulate human-like Theory of Mind (ToM) to assist with routine tasks. However, existing methods for evaluating machine ToM focus primarily on unimodal models and largely treat these models as black boxes, lacking an interpretative exploration of their internal mechanisms. In response, this study adopts an approach based on internal mechanisms to provide an interpretability-driven assessment of ToM in multimodal large language models (MLLMs). Specifically, we first construct a multimodal ToM test dataset, GridToM, which incorporates diverse belief testing tasks and perceptual information from multiple perspectives. Next, our analysis shows that attention heads in multimodal large models can distinguish cognitive information across perspectives, providing evidence of ToM capabilities. Furthermore, we present a lightweight, training-free approach that significantly enhances the model's exhibited ToM by adjusting in the direction of the attention head. [1]

## 1. Introduction

Theory of Mind (ToM) is a psychological term referring to infer mental states to self and others. This capability is fundamental to human social cognition and emotional understanding. In recent years, the rapid development of large models raised researchers' consideration: can they interact with us in a manner similar to humans? For example, could

they, like the robot TARS in Interstellar, accurately comprehend and execute both explicit and implicit tasks assigned by humans (Figure 1) ? It leds to the question that whether large models have ToM. Evaluating the ToM capabilities of large models is crucial for understanding their potential to meaningfully engage with human communication and reasoning. Some works utilized the classical Sally-Anne to test the ToM capabilities of machines (Nematzadeh et al., 2018; Grant et al., 2017; Ullman, 2023; Lore et al., 2024; Kosinski, 2024). While these evaluation methods offer preliminary insights into the ToM capabilities of large models, they remain limited in scope.

Most existing studies adopt unimodal approaches, focusing on either text or videos, and lack comprehensive agent-level information (Gandhi et al., 2021; Nematzadeh et al., 2018; Grant et al., 2017; Le et al., 2019; Amirizaniani et al., 2024). In contrast, human social interactions rely on reasoning about others' mental states by integrating multimodal inputs, such as visual and linguistic data. Although some studies (Jin et al., 2024; Shi et al., 2025) have attempted to extend ToM evaluations of large models to multimodal environments using video-based datasets, their datasets often incorporate excessive high-level information, such as spatial relationships, agents' tasks, and action trajectories (Ma et al., 2023). Moreover, in real-world datasets, an agent's perception of environmental events cannot be accurately captured. For example, in the MMToM-QA dataset (Jin et al., 2024), it is impossible to determine from the video modality alone whether the protagonist truly "saw" the plate. Consequently, the accuracy of ToM evaluations may depend on the quality of perceptual information, which could lead to correct or incorrect performance for reasons unrelated to genuine ToM capabilities. Unlike these prior works, we construct a dataset that is based on 2d grid world, enabling large models to perceive the full context of the physical world through the video modality while supplementing cognitive perspective information for each agent through the text modality.

Furthermore, the majority of assessments of ToM capabilities in large models take a black-box approach, relying heavily on question-answering tasks to infer conclusions (Xu et al., 2024), as demonstrated in Figure 1, while lacking

---
[*]Equal contribution  [1]School of Computer and Communication Engineering, University of Science and Technology Beijing, Beijing, China. Correspondence to: Bochao Zou <zoubochao@ustb.edu.cn>, Huimin Ma <mhm-pub@ustb.edu.cn>.

*Proceedings of the $42^{nd}$ International Conference on Machine Learning*, Vancouver, Canada. PMLR 267, 2025. Copyright 2025 by the author(s).

[1]Project Page: https://annaisavailable.github.io/GridToM

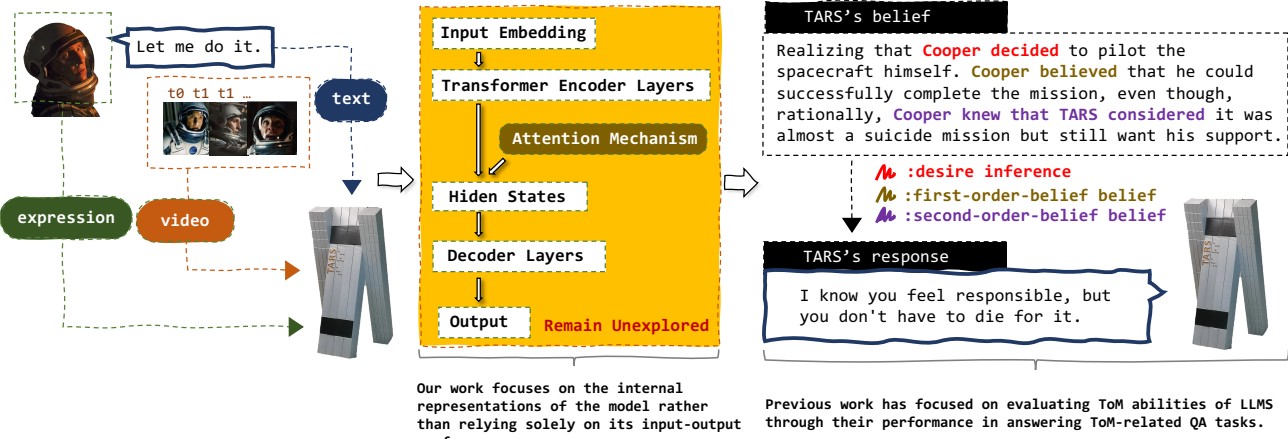

*Figure 1.* This illustration highlights the integration of different levels of ToM: recognizing an agent's desire (Cooper wants to pilot), a first-order belief (he believes he can do it), and a second-order belief (he believes TARS perceives it as risky). These nested mental states are crucial in evaluating advanced ToM.

interpretability-oriented methodologies (Mao et al., 2024). However, multimodal large language models (MLLMs) are known to exhibit hallucination phenomena, where the quality of prompts can significantly impact their performance on question-answering tasks. This means that a model may "understand" a concept but fail to provide a "correct" response (Bai et al., 2024). Like demonstreted in Figure 2, factors influencing QA performance are not limited to ToM capabilities. Elements such as hallucination and scenario understanding also affect the ToM evaluation results of previous studies. Consequently, it is insufficient to determine whether MLLMs possess ToM capabilities solely based on their performance in ToM tasks. In contrast, our goal is to examine whether these models develop internal representations that distinguish agents' mental states from different perspectives, beyond merely analyzing output accuracy. This will provide an interpretable explanation of whether MLLMs possess ToM capabilities.

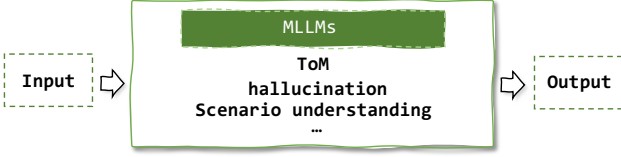

*Figure 2.* The figure highlights the limitations of current ToM evaluations, namely that other model capabilities (such as hallucination and scenario understanding) may interfere with the results.

In summary, our main contributions are as follows: (1) we introduce GridToM, a novel multimodal ToM dataset that in-

corporates diverse belief-testing tasks alongside perceptual information from multiple perspectives; (2) we conduct an in-depth analysis of the internal representations of MLLMs through interpretability methods, focusing on their intermediate activations; (3) we propose a training-free approach to enhance ToM performance in MLLMs by strategically shifting activations along specific directions.

## 2. Related Work

### 2.1. Dataset for Evaluating Theory of Mind

Inspired by traditional experiments used to evaluate ToM in children, some studies have applied the classic Sally-Anne task to assess ToM capabilities in machines (Grant et al., 2017; Eysenbach et al., 2016; van Duijn et al., 2023). Most existing datasets and methods for evaluating ToM capabilities are based on a single modality (Xiao et al.; Amirizaniani et al., 2024; Wu et al., 2023; Yim et al., 2024) . For text inputs, Mindgames (Sileo & Lernould, 2023) is a dataset grounded in dynamic epistemic modal logic, designed to evaluate the epistemic reasoning of large language models through controlled problem generation. OpenToM (Xu et al., 2024) is a dataset characterized by long-form narratives featuring real-world individuals and events, emphasizing the complexity of storylines and the diversity of character relationships. Similarly, ToMi (Le et al., 2019) is a comprehensive dataset encompassing multi-agent scenarios, multi-episode contexts, multi-turn question answering, and tasks involving mental state reasoning. Some tried videos, (Shu et al., 2021) is a benchmark composed of programmatically generated 3D animations, where agents interact with objects and move within various physical constraints.

SymmToM (Sclar et al., 2022) is a multi-agent reinforcement learning environment called SymmToM, where agents can simulate the mental states of others.

The MMToM-QA dataset (Jin et al., 2024) is the pioneering resource aimed at assessing machine learning models' ability to infer mental states from multimodal data, combining video and text in real-world tasks. Similarly, other studies (Chen et al., 2024; Shi et al., 2025) have also explored this domain. However, real-world video datasets often lack perspective information, making it challenging to infer high-dimensional details such as whether the protagonist in a story truly notices specific objects. This limitation can affect the accuracy of ToM task performance.

To address these challenges, we developed a dataset based on a 2D grid world environment, which provides simplified character relationships, complete physical information, and comprehensive perceptual data for all agents. The 2D grid world framework not only enables the creation of manipulable visual causal stories for training classifiers to distinguish perspective information but also avoids introducing high-level information. This reduces the cognitive burden on MLLMs, allowing them to focus on the core ToM tasks.

## 2.2. Benchmark

The question of whether large models exhibit genuine ToM capabilities remains a topic of ongoing debate. Some evaluation studies suggest that certain large models demonstrate a degree of ToM ability in reasoning tasks, such as understanding others' beliefs, intentions, and mental states (Kosinski, 2023; Bubeck et al., 2023; Zhou et al., 2023). However, other studies argue that the observed ToM-like capabilities of large models are not based on true generalization but instead result from learning patterns in question-answering tasks (Shapira et al., 2024; Ullman, 2023; Strachan et al., 2024) or lack of ToM (Sap et al., 2022; Verma et al., 2024).Most conclusions about ToM capabilities in large models rely on performance in QA tasks. In contrast, our research aims to address this question from an interpretability perspective by investigating the internal representations of MLLMs related to mental state understanding, rather than solely depending on the quality of their question-answering performance.

## 3. GridToM

**Why not previous grid world based ToM dataset?** The previous datasets only included unimodal inputs and lacked annotations for character perspective information and event details, making them unsuitable as positive and negative samples for the subsequent experiments in this study.

Unlike previous ToM works, GridToM provides manipulable multimodal visual-linguistic causal stories and includes the perceptual information of all agents in the scene. For each story, we apply randomized manipulations to the evaluation data, including room configurations, agent states, and action trajectories.

## 3.1. Overview

GridToM is generated based on the Multigrid library (Oguntola et al., 2023), which builds on Minigrid (Chevalier-Boisvert et al., 2023). It provides a multi-agent discrete gridworld environment, a simple and commonly used setting for ToM research in the machine learning community. The complete dataset construction pipeline and accompanying quality-control procedures are detailed in Appendix I. It has been demonstrated that a simple 2D gridworld can effectively support the development of diverse ToM tests (Ma et al., 2023), encompassing all mental states defined in ATOMS (Abilities in ToM Space) (Beaudoin et al., 2020).

Our dataset comprises 1,296 video-text pairs, with each video having a resolution of 294×420 pixels and approximately 40 frames. Each map is a 10×7 grid featuring three rooms and two agents. The dataset includes 27 distinct maps, each with two initial agent positions, two orientations, six sequences of agent movements into target rooms, and paired True Belief (TB) and False Belief (FB) stories, generating the 1,296 pairs. An example is shown in Figure 3.

## 3.2. Baseline

Our experiments reproduce the classic unexpected transfer task (Baron-Cohen et al., 1985). We simulate a complete interaction process within the gridworld environment. The testing dataset consists of 500 samples. An additional 148 samples were used for training and validation of our model, with 75% allocated for training and 25% for validation. To ensure that each selected model processes the input without errors, we use four key frames and three intermediate frames between them as input for video-based tasks, instead of providing all frames. The temperature of all models was set to 0.

**Human Participants** To evaluate human performance in the proposed dataset, 12 human participants were recruited to answer the test questions, all of them gave informed consent. The age range was from 23 to 32 years, with an average of 24.8 (SD = 2.3). Each participant was randomly assigned 100 samples from the 500-test dataset, and the final performance score was obtained by averaging their results. Importantly, in the video-only condition, core environmental rules—including that closed doors block agents' perception—were clearly explained prior to testing. Any omitted narrative content was non-instructional and did not impact task comprehension. This ensures that participants were not misled or disadvantaged by the absence of textual guidance.

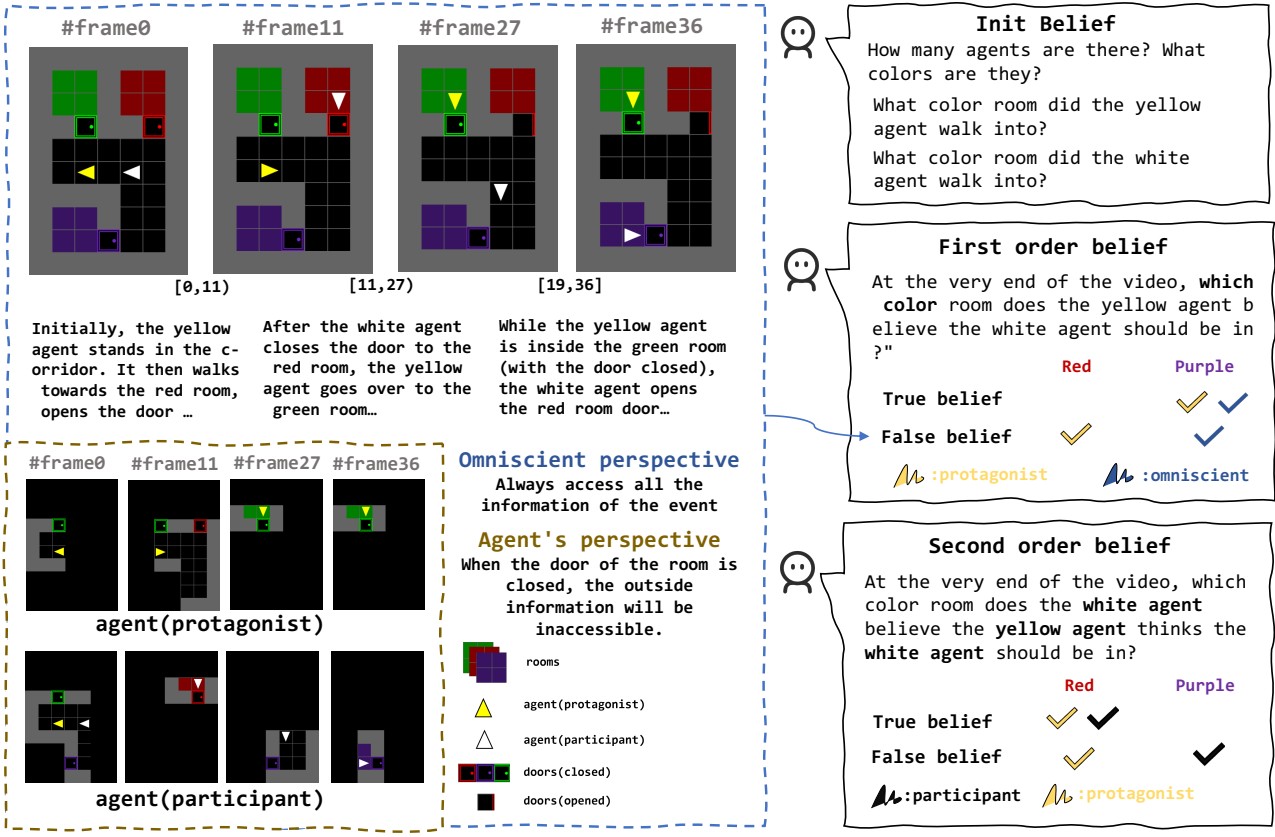

*Figure 3.* A FB sample in GridToM includes an omniscient-perspective video covering the entire event timeline, along with omniscient-perspective textual descriptions for each time interval. For all agents involved in the event, we provide full physical perspective information across the timeline. When an agent closes a door, we mask its perception of any information beyond the door to simulate realistic sensory limitations. Each sample contains three types of questions (illustrated on the right). For each video-text pair, the accompanying text annotations include environment descriptions, initial belief assessments, first-order belief assessments, and second-order belief assessments. We provide the full text and the video in Appendix C.

**MLLMs** We evaluated MLLMs under both multimodal and pure-video conditions, including GPT-4o (Achiam et al., 2023), Doubao-1.5-vision-pro (Team, 2025), DeepSeek-vl2-small (Liu et al., 2024), LLava-Next-Video-7b-hf (Touvron et al., 2023), Qwen2-VL-7b-instruct (Bai et al., 2023). Additionally, for fairness, we evaluated both subclasses of MLLMs using the same methodology, following previous work (Jin et al., 2024). Specifically, we sample a fixed set of seven frames from the video (including four key frames and three evenly sampled intermediate frames) along with the corresponding annotations to evaluate Image-Text-to-Text Models and Video-Text-to-Text Models.

**LLMs** We also evaluated GridToM on various large language models (LLMs) using text-based input only, including GPT-4o (Achiam et al., 2023), Doubao-1.5-Pro-32k (Team, 2025), DeepSeek-V3 (Liu et al., 2024), LLaMA-3.3-70B-Instruct (Dubey et al., 2024), Mistral-7B-Instruct-V3 (Jiang et al., 2023), LLaVA-Next-Video-7B-HF (Touvron et al.,

2023), and Qwen-VL-7B-Instruct (Bai et al., 2023).

We evaluate the models under three conditions (Jin et al., 2024): Multimodal QA with both video and text inputs, Text-only QA with text inputs only, and Video-only QA with video inputs only. We list our detailed setting in Appendix F. Results are in Table 1, we demonstrate baseline of existing MLLMs on GirdToM, while the result of initial belief test is in Appendix D.

## 4. Belief representation in MLLMs

### 4.1. Model

In the exploration and modification phases, we utilize the LLaVA-Next-Video model, a MLLM specifically designed for video understanding and generation tasks. Additionally, we utilized the Qwen2-VL model to perform the aforementioned two phases, demonstrating the effectiveness of our

approach.

## 4.2. Attention Feature Extraction

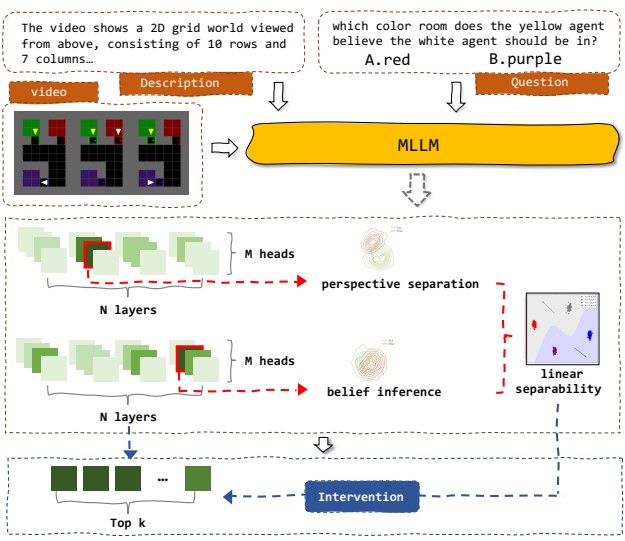

*Figure 4.* Overview of Our Workflow. We first constructed the GridToM dataset and conducted benchmark testing of MMLMs on it. Subsequently, we input video-text pairs to probe the internal attention representations of the models. Using logistic regression, we performed binary classification on the representations of positive and negative samples to identify attention heads that are sensitive to perspective separation and belief representation. Targeted interventions were then applied to the top $K$ most sensitive attention heads during inference.

We begin by investigating whether MLLMs represent and how they represent the beliefs of different agents. Our objective is to decode the belief states of various agents from the activations of attention heads, given multimodal story narratives and corresponding belief statements.

Specifically, MLLMs first embed the input multimodal data into high-dimensional spaces, including visual inputs $V = \{v_1, v_2, ..., v_m\}$ and textual inputs $X = \{x_1, x_2, ..., x_n\}$, where $m$ and $n$ represent the token lengths of the visual and textual inputs, respectively. The model concatenates the visual and textual embeddings into a unified input sequence $T = concat(V, X) \in \mathbb{R}^{(m+n) \times DH}$, where $D$ denotes the dimension of each attention head and $H$ represents the number of attention heads. This unified input is then passed through a Transformer architecture with $L$ layers.

At each layer, the concatenated input undergoes multi-head attention. The multi-head attention mechanism (MHA) can be approximated as Equation (1):

$$T_{l+1} = T_l + \sum_{h=1}^{H} Attn_l^h(P_l^h T_l) \cdot W_l^o, \tag{1}$$

Where $Attn_l^h$ denotes the attention operation of the $n$-th head at the $l$-th layer, $P_l^h \in \mathbb{R}^{D \times DH}$ maps stream activation into a $D$-dimensional head space, and $W_l^o \in \mathbb{R}^{D \times DH}$ is the output projection matrix. Inspired by (Li et al., 2024), the probing and intervention steps occur after $Attn$ and before $W$.

We extract the output of each attention head at every layer, capturing the activation at the final token position, denoted as $X \in \mathbb{R}^{L \times H \times D}$. Each attention head activation is associated with belief labels $Y_p$ and $Y_o$ , which represent the correctness of the protagonist's perspective and the omniscient perspective, respectively.

Due to the simplicity of the 2D gridworld, in TB scenarios, the protagonist's perceptual information is equivalent to omniscient information. This allows the protagonist's perspective video to be substituted by the omniscient perspective video. In TB scenarios, the protagonist's belief labels $Y_p = True$ and $Y_p = False$ help identify the layers and heads sensitive to reasoning based on perceptual information. In FB scenarios, the protagonist's belief labels help identify the layers and heads that are sensitive to integrating belief information across perspectives. For the omniscient belief label $Y_o$, the correct label corresponds to multimodal data with an omniscient perspective and accurate belief inference, while the incorrect label includes either an incorrect perspective or an incorrect inference result.

This design of correct and incorrect labels targets two aspects: perspective separation and belief inference, integrating them into a unified framework. Targeted interventions are applied to the heads that are sensitive to these two aspects. We collectively define correct perspective separation and correct belief inference as true labels, and their opposites as false labels. In our approach, we only use the correct and incorrect labels from the protagonist's perspective to indicate and guide perspective separation and belief reasoning.

$$Y_p = \{Y_p^{TB} \cap Y_p^{FB}\} \tag{2}$$

For different belief tasks, our probing strategies vary slightly, while the interference strategy remains consistent, as detailed in Appendix B.

### 4.3. Probing

Probe is a standard tool for analyzing the internal representations of networks (Köhn, 2015; Gupta et al., 2015). The idea is to train a classifier (probe) on the activations of the network to distinguish specific types of inputs or outputs.

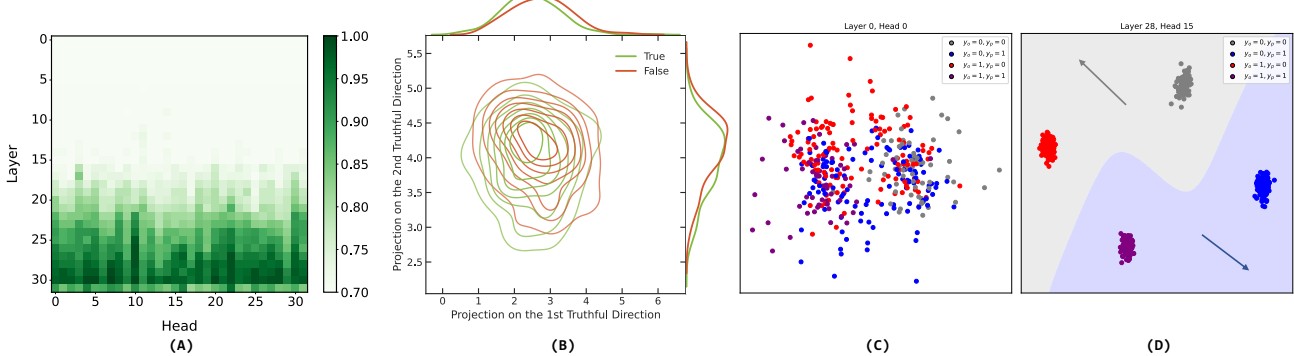

*Figure 5.* (A) The linear probing accuracy of all heads across all layers in LLaVA-Next-Video on the test set. The x-axis represents the heads, and the y-axis represents the layers. Dark green indicates higher accuracy, with 50% serving as the baseline accuracy for random guessing. (B) Kernel density estimation (KDE) plot of activations in layer 28, head 15 of LLaVA-Next-Video, projected onto the top two true directions, showing real (green) and false (orange) pairs. Marginal distributions are displayed along the top and right axes. (C) & (D) The linear separability of belief representations is explained through a visual interpretation of the typical representation space, demonstrating the attention feature extraction strategy proposed in Appendix B. The binary combinations of $Y_p$ and $Y_o$ labels correspond to the combinations of TB and FB with correct and false beliefs. For belief-sensitive heads (e.g., head 15 in layer 28), they can effectively estimate the boundaries of the belief states for both the omniscient perspective and the protagonist's perspective, whereas insensitive heads cannot. These four combinations form distinct clusters in the representation space without overlap, with clearly defined decision boundaries. The probing weight direction represents the decision boundary, effectively separating these belief combinations.

$$f_l^h = \frac{1}{1 + e^{-(x\theta + b)}}, \qquad (3)$$

where $f_l^h$ denotes a logistic sigmoid function for $(x\theta + b)$, while $\theta \in \mathbb{R}^D$ and $b \in \mathbb{R}$ represent the weight vector and bias, respectively. The parameters $\theta$ and $b$ are optimized by minimizing the cross-entropy loss.

We first conducted probing experiments on GridToM. The results are shown in Figure 5. The probing results for different models are listed in Appendix G. Subsequently, we performed the same probing experiments on the real-world multimodal ToM dataset MMToM-QA, to validate the generalizability of our probing method. The detailed information of the MMToM-QA dataset is presented in Appendix H, as shown in Figure 22.

For each attention head at every layer, we train a separate linear binary probe to fit the belief labels $Y_p$ and $Y_o$. Given a dataset of size $N$, we obtain the corresponding activations of a single attention head, denoted as $X \in \mathbb{R}^{N \times D}$, the corresponding belief labels are $Y \in \{0, 1\}^N$. We use a logistic regression model to predict the probability of the belief being true. In simple terms, we select the top $K$ attention heads ranked by accuracy in Figure 5 (A) and use the decision boundary in Figure 5 (C) as the direction for intervention weights. Figure 5 (A) shows the validation accuracy of the linear probe, indicating that many attention heads can accurately capture belief states from the protagonist's perspective. These abundant informational representations are

distributed across different heads in various layers, starting from middle layers to the final layers, whereas the initial layers lack this capability.

Meanwhile, Figure 5 (C) demonstrates the linear separability of belief representations. We visualize the attention feature extraction strategy proposed in Appendix B, where the four clusters represent the correctness of perspective separation and belief inference. These four combinations are distinctly clustered without overlap, with clear decision boundaries. This suggests that MLLMs indeed develop intermediate representations reflecting multi-perspective information extraction and belief inference based on the complete information provided. This phenomenon indicates that these attention heads implicitly encode the belief states of other perspectives in a linearly decodable manner. Furthermore, due to the simplified information in the 2D grid world, these implicit beliefs are easily propagated to the final layers.

To further understand belief representations in the activation space of attention heads, we visualized the geometric shapes within the activation space, as shown in Figure 5(b). Specifically, we reduced the activation space dimensions to two using Principal Component Analysis and selected two orthogonal directions ($\theta \perp \theta'$) with the maximum variance for separating true and false features. We visualized the geometric projections onto $\theta$ and $\theta'$, observing partial overlap and distinct representations between the two distributions. Notably, the second direction still exhibits unique representation distributions, suggesting that the concepts of "true"

and "false" coexist in subspaces within the attention space, rather than being confined to a single unified space.

## 4.4. Intervention

Although the probing results have demonstrated that MLLMs possess internal mental representations, we still aim to intervene with the attention activation heads to further validate the practical significance of the classifier's directional representations during probing. Due to dataset limitations, MMToM-QA can only provide positive and negative samples in the text modality rather than multimodal ones, so we performed the intervention experiments exclusively on GridToM.

We first select the top $K$ attention heads with the highest sensitivity on the validation set, representing those most responsive to the differences between true and false beliefs. We then intervene on these selected heads after multi-head attention computation but before mapping back to the output, as computed as follows:

$$T_{l+1} = T_l + \sum_{h=1}^{H} (Attn_l^h(P_l^h T_l) + \alpha \sigma_l^h \theta_l^h) \cdot W_l^o, \quad (4)$$

where $\sigma_l^h$ denotes the standard deviation of activations along the target direction, and $\theta_l^h$ represents the intervention target direction, derived from the weight vector of the selected attention head. The parameter $\alpha$ controls the strength of the intervention. For the selected $K$ heads, $\alpha$ scales the activation along the original direction by $\alpha$ times the standard error in the target direction.

We provide an analysis of the effects of hyperparameters $K$ and $\alpha$ on the intervention results in the Appendix E. The analysis shows that our approach relies on an interpretable intervention based on internal representations and input perturbations, rather than on hyper-parameter tuning. By identifying the attention heads responsible for true belief representation and applying targeted interventions, we enhance the sensitivity of the model to perspective separation and belief representation. This ultimately improves the MLLM's ability to perceive and represent beliefs more effectively.

## 5. Experiments

### 5.1. Result

We present a summary of our results in Table 1. For each task, we include human accuracy as a benchmark to represent the upper bound of task performance.

In the multimodal setting, humans achieved high accuracy across TB, FB, and Both conditions, demonstrating the consistency of our design. However, in video-only tasks,

performance on TB tasks declined slightly, as humans inferred the protagonist's perspective but were occasionally misled by scenarios contradicting physical intuition, such as omniscient visibility through a doorway. The absence of textual clarification further amplified these misjudgments, as prior knowledge influenced their reasoning. Similarly, in the text-only setting, human performance experienced a slight decline due to excessive textual interference, which introduced confusion and contributed to errors.

In first-order belief task, the baseline results in the multimodal setting indicate that MLLMs achieve high accuracy on FB tasks (e.g., both ChatGPT-4.0 and Doubao-1.5-Vision-Pro reach 100%), even slightly surpassing human performance (99.9%). However, their performance on TB tasks is significantly weaker (e.g., ChatGPT-4.0 achieves 6.2%, Doubao-1.5-Vision-Pro achieves 16.8%, and 0% on second-order belief tasks). We attribute this discrepancy to the models' overreliance on patterns learned from FB tasks, which may lead to misgeneralization in TB scenarios. This sensitivity prevents MLLMs from recognizing critical contextual details, such as the fact that the protagonist's door is open in TB tasks. This inference is supported by the following observations: When the influence of visual factors related to physical spatial positions is removed, LLMs (e.g., ChatGPT-4.0, Llama) still perform poorly when processing text-only inputs. However, MLLMs (e.g., ChatGPT-4.0, LLava-Next-Video, and Qwen2-VL) demonstrate better performance when presented with pure video containing physical spatial information (excluding textual influences). This highlights the importance of establishing reasonable reasoning processes in both visual and textual modeling to balance task performance.

In both multimodal and video-only conditions, the poor performance on TB tasks negatively impacts all MLLMs' performance on Both tasks (i.e., correctly answering both TB and FB tasks for the same set). The performance on Both tasks provides an intuitive reflection of MLLMs' ability to handle belief reasoning tasks; high accuracy on a single task may indicate excessive sensitivity rather than a balanced reasoning capability. Under the text-only condition, LLMs (e.g., ChatGPT-4.0, Doubao, and Deepseek) also exhibit relatively high accuracy on TB tasks. Interestingly, Doubao-1.5-Pro-32k stands out by achieving 100% accuracy on both tasks. In second-order belief tasks, MLLMs perform near the random guessing baseline (50%) and struggle on the Both task, highlighting the challenge. In contrast, LLMs excel in text-only tasks.

Table 1 also presents the results of applying our activation interference strategy to two MLLMs. While our attention feature extraction strategies are slightly adjusted for different belief tasks, the probing and interference methodology remains consistent, as detailed in Appendix B. The table

*Table 1.* Model performance comparison on the GridToM benchmark. TB = True Belief. FB = False Belief. For TB and FB, the expectation for random guesses is 50%. Both indicates a situation where both TB and FB are judged as correct for a given set.

| | METHOD | SETTING | FIRST-ORDER | | | SECOND-ORDER | | |
|---|---|---|---|---|---|---|---|---|
| | | | TB(%) | FB(%) | BOTH(%) | TB(%) | FB(%) | BOTH(%) |
| **MULTIMODAL** | HUMAN | | 99.9 | 99.9 | 99.8 | 99.9 | 99.8 | 99.8 |
| | CHATGPT 4O | BASELINE | 6.2 | 100.0 | 6.2 | 50.0 | 47.3 | 5.3 |
| | DOUBAO-1.5-VISION-PRO | | 16.8 | 100.0 | 16.8 | 50.0 | 45.2 | 17.1 |
| | DEEPSEEK-VL2-SMALL | | 68.4 | 43.8 | 13.4 | 56.9 | 46.8 | 7.0 |
| | LLAVA-NEXT-VIDEO-7B-HF | | 53.2 | 42.8 | 0.8 | 48.7 | 39.8 | 0.2 |
| | QWEN2-VL-7B-INSTRUCT | | 26.6 | 97.0 | 23.6 | 64.3 | 37.8 | 15.4 |
| | LLAVA-NEXT-VIDEO-7B-HF | +α | 63.8(+10.6) | 51.6(+8.8) | 22.0(+21.2) | 61.1(+12.4) | 42.2(+2.4) | 10.2(+10.0) |
| | QWEN2-VL-7B-INSTRUCT | +α | 60.4(+33.8) | 97.4(+0.4) | 31.2(+7.6) | 75.1(+10.8) | 46.6(+8.8) | 24.2(+8.8) |
| **VIDEO** | HUMAN | | 84.6 | 99.8 | 84.6 | 81.0 | 99.8 | 81.0 |
| | CHATGPT 4O | BASELINE | 69.6 | 30.4 | 5.6 | 49.2 | 37.8 | 3.4 |
| | DOUBAO-1.5-VISION-PRO | | 46.6 | 55.8 | 7.2 | 50.6 | 42.0 | 1.1 |
| | DEEPSEEK-VL2-SMALL | | 55.6 | 44.0 | 2.8 | 48.9 | 48.4 | 3.7 |
| | LLAVA-NEXT-VIDEO-7B-HF | | 50.8 | 48.2 | 0.4 | 50.1 | 41.2 | 0.3 |
| | QWEN2-VL-7B-INSTRUCT | | 52.2 | 48.6 | 5.2 | 49.1 | 40.0 | 2.1 |
| | LLAVA-NEXT-VIDEO-7B-HF | +α | 54.4(+3.6) | 51.2(+3.0) | 16.4(+16.0) | 55.9 (+5.8) | 42.2(+1.0) | 12.3(+12.0) |
| | QWEN2-VL-7B-INSTRUCT | +α | 53.8(+1.6) | 52.2(+3.6) | 18.6(+13.4) | 52.1(+3.0) | 46.0(+6.0) | 19.5(+17.4) |
| **TEXT** | HUMAN | | 98.0 | 98.1 | 96.6 | 97.6 | 98.0 | 96.3 |
| | CHATGPT 4O | BASELINE | 14.2 | 100.0 | 14.2 | 50.0 | 72.3 | 39.9 |
| | DOUBAO-1.5-PRO-32K | | 100.0 | 100.0 | 100.0 | 75.6 | 71.1 | 50.9 |
| | DEEPSEEK-V3 | | 84.4 | 100.0 | 84.4 | 61.5 | 70.9 | 41.4 |
| | LLAMA-3.3-70B-INSTRUCT | | 0.0 | 100.0 | 0.0 | 50.7 | 70.3 | 23.7 |
| | MISTRAL-7B-INSTRUCT-V0.3 | | 77.8 | 47.6 | 25.4 | 62.9 | 39.9 | 14.5 |
| | LLAVA-NEXT-VIDEO-7B-HF | | 40.8 | 56.4 | 0.0 | 49.3 | 50.1 | 0.7 |
| | QWEN2-VL-7B-INSTRUCT | | 48.6 | 66.2 | 14.8 | 61.4 | 52.7 | 16.3 |

highlights that our method effectively modifies the models' behavior, resulting in substantial performance improvements across first-order and second-order belief tasks under multimodal conditions, including TB, FB, and Both.

Additionally, in Figures 15 and 16 of Appendix E, we illustrate the impact of hyperparameters on the interference effect. Specifically, the weight direction of the probed protagonist's perspective has a significant impact on baseline performance, highlighting its critical role in the ToM reasoning process. As expected, steering the reasoning direction of MLLMs toward this perspective consistently improves the accuracy of TB and FB tasks. Throughout this process, no invalid responses are generated until the maximum value is reached, at which point all responses become invalid. We also tested interference directed toward the omniscient perspective. Due to the differing effects of perspective separation, its interference effect was observed to be lower than that of the protagonist's perspective. This finding aligns with our expectations, further confirming the importance of correctly aligning the models' reasoning direction with the protagonist's perspective for improved task performance.

## 5.2. Discussion

In this study, we introduced GridToM, a novel multimodal dataset characterized by its incorporation of diverse belief-testing tasks and perceptual information from multiple perspectives. Designed to evaluate the ToM capabilities of MLLMs, this dataset enables comprehensive assessments of their reasoning abilities across varied scenarios. We conducted comprehensive tests of existing MLLMs on this dataset. We observed that these models perform better on text-based data compared to video data. While the ToM capabilities exhibited in multimodal settings may be less pronounced than in unimodal scenarios, real-world applications, such as real-time human-machine collaboration, often necessitate multimodal data inputs. Moreover, in such contexts, the feasibility of providing purely textual input in real-time is limited, emphasizing the necessity of evaluating ToM capabilities and interpretability in MLLMs.

Through analysis of MLLMs' internal mechanisms, we identified attention heads capable of distinguishing different perspective information and reasoning about correct beliefs. By modifying the reasoning attention direction based on the activation direction indicated by these attention heads, we achieved significant enhancement of ToM capabilities

in both first-order and second-order belief tasks, further validating the effectiveness of this mechanism.

However, our study has certain limitations. First, the tasks in our dataset are limited to first-order and second-order belief tasks within the ATOMs framework (Beaudoin et al., 2020), whereas ToM theory encompasses a broader range of tasks that remain unexplored. Second, due to restrictions in accessing model code, our approach was only validated on a limited selection of MLLMs.

## Acknowledgements

This work was supported by the National Science and Technology Major Project (2022ZD0117902, 2022ZD0117901) and the the National Natural Science Foundation of China (No. 62206015, 62227801, 62376024). We thank the anonymous reviewers for insightful discussions.

## Impact Statement

Understanding human mental states is crucial for developing AI that interacts effectively and empathetically. Our benchmark advances ToM evaluation in MLLMs by integrating belief-testing tasks and interpretability analysis, revealing AI cognitive mechanisms. Grounded in cognitive science, it prioritizes fairness, inclusivity, and invites community feedback to refine human-aligned AI systems.

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

# Appendix

## A. Benchmark Details

*Table 2.* A comparison of Theory of Mind benchmarks (1s and 2nd belief tasks).

| DATASET | ToM ASPECT | TESTED CONCEPTS | TEST SIZE | MODALITY | PERCEPTUAL INFORMATION |
|---|---|---|---|---|---|
| ToMi(Le et al., 2019) | 1st & 2nd | FB | 400 | TEXT | NO |
| MINDGAMES(Sileo & Lernould, 2023) | 1st & 2nd | FB | 400 | TEXT | NO |
| ADV-CSFB(Kosinski, 2023) | 1st & 2nd | FB & TB | 183 | TEXT | NO |
| HI-TOM(Wu et al., 2023) | 1st | FB & TB | 600 | TEXT | NO |
| MMToM-QA(Jin et al., 2024) | 1st | FB & TB | 600 | TEXT & VIDEO | NO |
| GridToM(Ours) | 1st & 2nd | FB & TB | 1296 | TEXT & VIDEO | YES |

## B. Attention feature extraction strategies

### B.1. First-order Belief

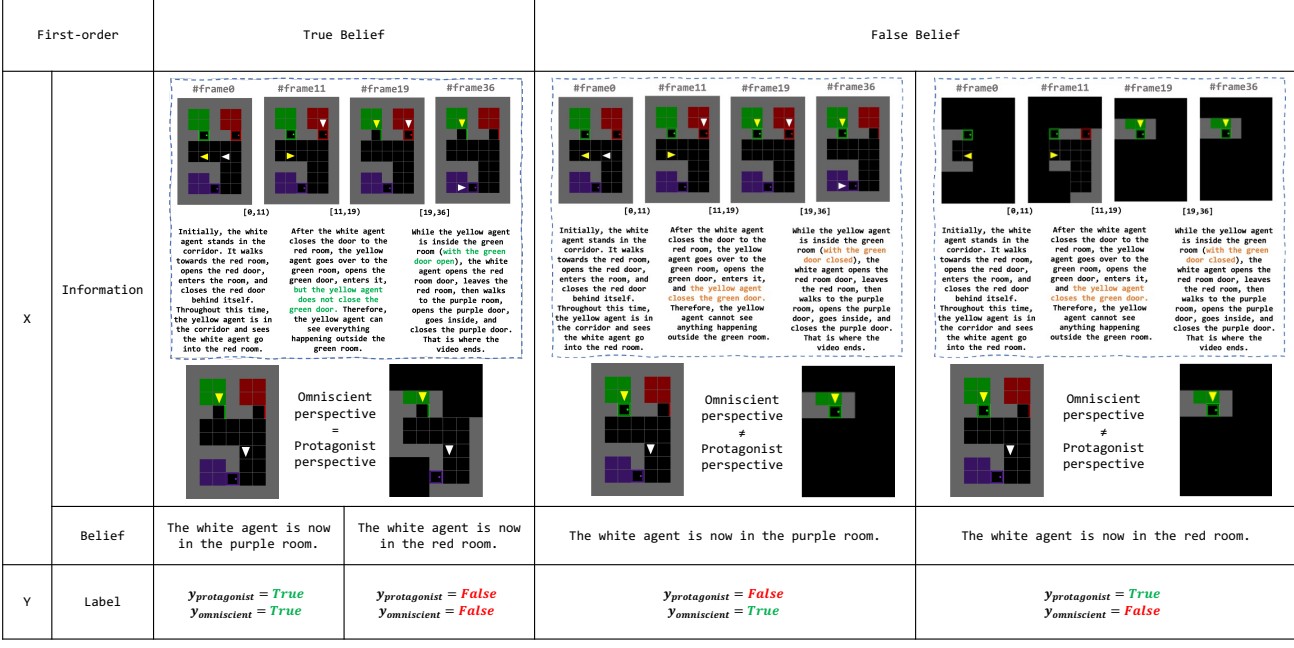

*Figure 6.* In the attention feature extraction process for first-order belief tasks, the information obtained from the omniscient and protagonist perspectives is consistent in the TB task. We identify belief-reasoning-sensitive features in attention by comparing correct and incorrect belief pairs. However, in the FB task, the protagonist's perspective has limited information. Therefore, we use the visual information from the protagonist's perspective along with the corresponding annotations as positive samples, while the omniscient perspective serves as negative samples. By comparing positive and negative samples, we identify attention features sensitive to perspective separation.

## B.2. Second-order Belief

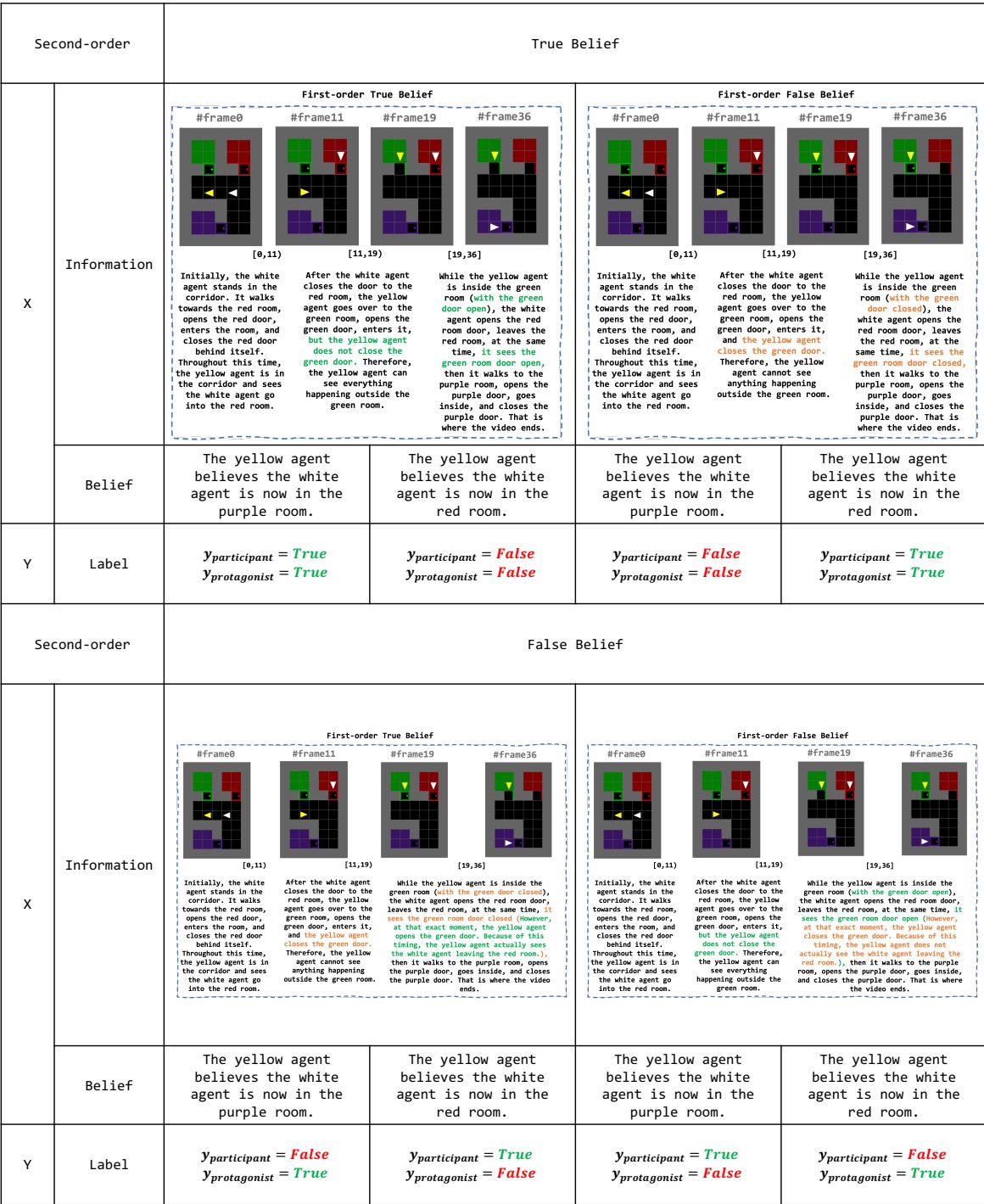

*Figure 7.* In the attention feature extraction process for second-order belief tasks, both the TB and FB tasks include the TB and FB tasks from first-order belief tasks. Unlike first-order belief tasks, the FB task in second-order belief reasoning contains the participant's incorrect perception of the protagonist's belief, achieved through a carefully designed timing setup. Since second-order belief reasoning involves the participant's belief about the protagonist's belief and does not include perspective separation tasks, we identify belief-reasoning-sensitive features in attention solely by comparing correct and incorrect belief pairs.

# C. Full Version of the Example Questions in Figure 3

## C.1. Videos

### TB test

The task of TB refers to the situation where the protagonist's beliefs align with those from an omniscient perspective, meaning the protagonist has access to all the information about the events. In the TB experiment, when the protagonist enters the room and leaves the door open, they are able to observe the situation outside the room, including the movements of the participants. We select a representative example from the dataset and present the video frame sequences from three distinct perspectives: the omniscient perspective (Figure 8, A), the protagonist's perspective (Figure 8, C), and the participant's perspective (Figure 8, B).

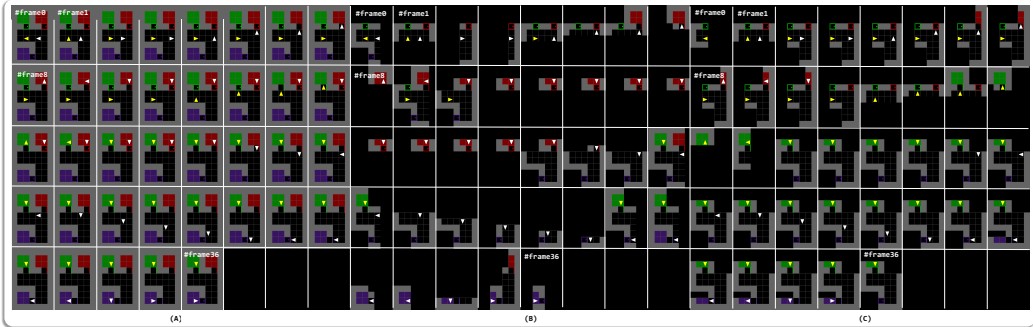

*Figure 8.* (A) The video frames from the omniscient perspective (36 frames in total) in TB test are shown in the figure. (B) The video frames from the participant's perspective (36 frames in total) in TB test are shown in the figure. (C) The video frames from the protagonist's perspective (36 frames in total) in TB test are shown in the figure.

### FB test

The task of FB refers to the situation where the protagonist's beliefs diverge from those of an omniscient perspective, meaning the protagonist does not have access to all the information about the events. In the FB experiment, the protagonist enters the room and does not observe critical events, such as the movements of the participants outside the room, due to the door being closed. We select a representative example from the dataset and present the corresponding video frame sequences from three distinct perspectives: the omniscient perspective (Figure 9, A), the protagonist's perspective (Figure 9, C), and the participant's perspective (Figure 9, B).

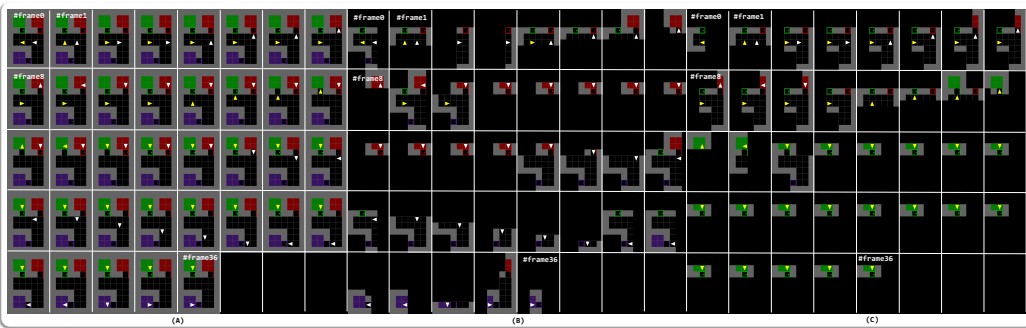

*Figure 9.* (A) The video frames from the omniscient perspective (36 frames in total) in FB test are shown in the figure. (B) The video frames from the participant's perspective (36 frames in total) in FB test are shown in the figure. (C) The video frames from the protagonist's perspective (36 frames in total) in FB test are shown in the figure.

## C.2. Text

### Initial Belief

The concept of initial belief refers to the foundational understanding or assumption MLLMs hold about the scenario before answering the ToM questions. In the context of this study, initial belief encompasses the MLLMs' pre-existing mental representation regarding three specific aspects of the task (see Figure 10):

- **Quantity and Color** A single question evaluates the agent's ability to interpret and reason about the numerical or visual attributes of objects based on its initial belief.

- **Spatial Understanding** Two questions assess the agent's capacity to comprehend and reason about the spatial arrangement or movement of objects within the environment.

This structured approach ensures that the evaluation effectively measures the agent's ToM capabilities within a multimodal framework.

---

**Spatial Location Information**

The video shows a 2D grid world viewed from above, consisting of 10 rows and 7 columns. Gray represents the wall and cannot be penetrated. Black squares with gray borders represent the corridor floor. There are three rooms in this grid world, each with its own color (and each room's door has the same color as the room). There are two triangles here, representing the agents.

**Initial Belief**

`"Question1":` How many agents (triangles) are there? What colors are they? Choose from the following colors and just answer the color(s). (white, green, red, yellow, purple)
`"section1":` [0,36]
`"answer1":` ["white", "yellow"]

`"Question2":` What color room did the white agent walk into? Choose from the following colors and just answer the color. (red, green, purple)
`"section2":` [0,11]
`"answer2":` ["red"]

`"Question3":` What color room did the yellow agent walk into? Choose from the following colors and just answer the color. (red, green, purple)
`"section3":` [0,19]
`"answer3":` ["green"]

---

*Figure 10.* The textual annotations for the initial belief task are shown in the figure.

### First-order belief

The concept of first-order belief refers to the direct inferences or reasoning that MLLMs make about the mental states of others, grounded in their observable actions or statements. To facilitate the subsequent training of classifiers and identifying the representational direction of perspective information, the dataset includes a single first-order belief question, along with two answer options and the correct answer. Additionally, the dataset provides the corresponding contents of the True TB and FB tests associated with the question. Furthermore, detailed descriptions of the story progression across different temporal segments are included to capture the sequence of events. This design ensures that the dataset not only facilitates the evaluation of first-order belief reasoning in MLLMs but also establishes a structured framework for identifying and analyzing perspective-based information through temporal and belief-based annotations (see Figure 11).

```
                                    First Order Belief
 True Belief
 "Question": At the very end of the video, which color room does the yellow agent believe the white agent should be in?
 "options": ["red", "purple"]
 "answer": "purple"
 "belief true": "The white agent is now in the purple room. "
 "belief false": "The white agent is now in the red room."
 "caption": "The story proceeds as follows: 1. Initially, the white agent stands in the corridor. It walks towards the red room, opens
    the red door, enters the room, and closes the red door behind itself. Throughout this time, the yellow agent is in the corridor and
    sees the white agent go into the red room. 2. After the white agent closes the door to the red room, the yellow agent goes over to the
    green room, opens the green door, enters it, but the yellow agent does not close the green door. Therefore, the yellow agent can see
    everything happening outside the green room. 3. While the yellow agent is inside the green room (with the green door open), the white
    agent opens the red room door, leaves the red room, then walks to the purple room, opens the purple door, goes inside, and closes the
    purple door. That is where the video ends."
 False Belief
 "Question": At the very end of the video, which color room does the yellow agent believe the white agent should be in?"
 "options": ["red", "purple"]
 "answer": "red"
 "belief true": "The white agent is now in the red room."
 "belief false": "The white agent is now in the purple room. "
 "caption": "The story proceeds as follows: 1. Initially, the white agent stands in the corridor. It walks towards the red room, opens
    the red door, enters the room, and closes the red door behind itself. Throughout this time, the yellow agent is in the corridor and
    sees the white agent go into the red room. 2. After the white agent closes the door to the red room, the yellow agent goes over to the
    green room, opens the green door, enters it, and the yellow agent closes the green door. Therefore, the yellow agent cannot see
    anything happening outside the green room. 3. While the yellow agent is inside the green room (with the green door closed), the white
    agent opens the red room door, leaves the red room, then walks to the purple room, opens the purple door, goes inside, and closes the
    purple door. That is where the video ends."
```

*Figure 11.* The textual annotations for the first order belief task in the TB and FB tests are shown in the figure.

**Second-order belief**

The concept of second-order belief pertains to the reasoning and inferences that MLLMs make regarding an agent's beliefs about another agent's mental state, based on observed actions or interactions. This evaluation also encompasses the question, answer options, the corresponding TB and FB conditions, as well as the story descriptions (see Figure 12 and Figure 13).

```
                                   Second Order Belief
 True Belief
 "Question": At the very end of the video, which color room does the white agent believe the yellow agent thinks the white agent should
 be in?
 "options": ["red", "purple"]
 "answer": "red"
 "belief true": "The yellow believes the white agent is now in the red room."
 "belief false": "The yellow believes the white agent is now in the purple room."
 "caption": "The story proceeds as follows: 1. Initially, the white agent stands in the corridor. It walks towards the red room, opens
 the red door, enters the room, and closes the red door behind itself. Throughout this time, the yellow agent is in the corridor and
 sees the white agent go into the red room. 2. After the white agent closes the door to the red room, the yellow agent goes over to the
 green room, opens the green door, enters it, but the yellow agent does not close the green door. Therefore, the yellow agent can see
 everything happening outside the green room. 3. While the yellow agent is inside the green room (with the green door open), the white
 agent opens the red room door, leaves the red room, at the same time, it sees the green room door open, then it walks to the purple
 room, opens the purple door, goes inside, and closes the purple door. That is where the video ends."

 "Question": At the very end of the video, which color room does the white agent believe the yellow agent thinks the white agent should
 be in?
 "options": ["red", "purple"]
 "answer": "purple"
 "belief true": "The yellow believes the white agent is now in the purple room."
 "belief false": "The yellow believes the white agent is now in the red room."
 "caption": "The story proceeds as follows: 1. Initially, the white agent stands in the corridor. It walks towards the red room, opens
 the red door, enters the room, and closes the red door behind itself. Throughout this time, the yellow agent is in the corridor and
 sees the white agent go into the red room. 2. After the white agent closes the door to the red room, the yellow agent goes over to the
 green room, opens the green door, enters it, and the yellow agent closes the green door. Therefore, the yellow agent cannot see
 anything happening outside the green room. 3. While the yellow agent is inside the green room (with the green door closed), the white
 agent opens the red room door, leaves the red room, at the same time, it sees the green room door closed, then it walks to the purple
 room, opens the purple door, goes inside, and closes the purple door. That is where the video ends."
```

*Figure 12.* The textual annotations for the second order belief task in the TB tests are shown in the figure.

```
                              Second Order Belief
False Belief
"Question": At the very end of the video, which color room does the white agent believe the yellow agent thinks the white agent should
be in?
"options": ["red", "purple"]
"answer": "red"
"belief true": "The yellow believes the white agent is now in the red room."
"belief false": "The yellow believes the white agent is now in the purple room."
"caption": "The story proceeds as follows: 1. Initially, the white agent stands in the corridor. It walks towards the red room, opens
the red door, enters the room, and closes the red door behind itself. Throughout this time, the yellow agent is in the corridor and
sees the white agent go into the red room. 2. After the white agent closes the door to the red room, the yellow agent goes over to the
green room, opens the green door, enters it, and the yellow agent closes the green door. Therefore, the yellow agent cannot see
anything happening outside the green room. 3. While the yellow agent is inside the green room (with the green door closed), the white
agent opens the red room door, leaves the red room, at the same time, it sees the green room door closed (However, at that exact moment,
the yellow agent opens the green door. Because of this timing, the yellow agent actually sees the white agent leaving the red room.),
then it walks to the purple room, opens the purple door, goes inside, and closes the purple door. That is where the video ends."

"Question": At the very end of the video, which color room does the white agent believe the yellow agent thinks the white agent should
be in?
"options": ["red", "purple"]
"answer": "purple"
"belief true": "The yellow believes the white agent is now in the purple room."
"belief false": "The yellow believes the white agent is now in the red room."
"caption": "The story proceeds as follows: 1. Initially, the white agent stands in the corridor. It walks towards the red room, opens
the red door, enters the room, and closes the red door behind itself. Throughout this time, the yellow agent is in the corridor and
sees the white agent go into the red room. 2. After the white agent closes the door to the red room, the yellow agent goes over to the
green room, opens the green door, enters it, but the yellow agent does not close the green door. Therefore, the yellow agent can see
everything happening outside the green room. 3. While the yellow agent is inside the green room (with the green door open), the white
agent opens the red room door, leaves the red room, at the same time, it sees the green room door open (However, at that exact moment,
the yellow agent closes the green door. Because of this timing, the yellow agent does not actually see the white agent leaving the red
room.), then it walks to the purple room, opens the purple door, goes inside, and closes the purple door. That is where the video
ends."
```

*Figure 13.* The textual annotations for the second order belief task in the FB tests are shown in the figure.

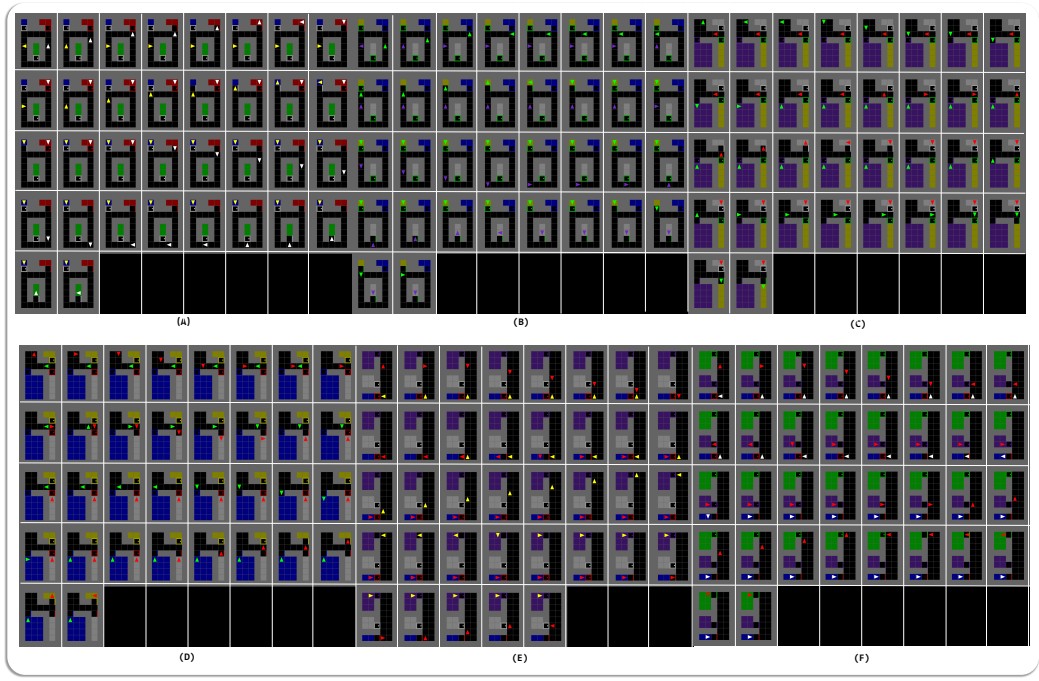

*Figure 14.* The figure presents video sequence frames extracted from three different rooms in the dataset as examples, where (A)(B), (C)(D), and (E)(F) correspond to different rooms. (A) and (B) illustrate examples of the same room configuration but with different agent states and action trajectories. Specifically, (A) represents the FB experiment, while (B) corresponds to the TB experiment. Similarly, (C) and (D) depict the FB and TB experiments, respectively, and (E) and (F) show the FB and TB experiments in another room configuration.

Furthermore, in our dataset, we apply randomized manipulations to the evaluation data for each story, including variations in room configurations, agent states, and action trajectories. This approach ensures diversity while preventing repetitive patterns that might result in spurious statistical correlations. To illustrate this, we have provided ten examples from the dataset, as shown in the Figure 14.

# D. Result of initial belief test in Section 3.2

### D.1. Result

We evaluated the initial belief accuracy (ACC%) of these MLLMs on the GridToM dataset. The results are shown in the table below (Table 3).

*Table 3.* Initial Belief

| | MODEL | ACC(%) |
|---|---|---|
| | HUMAN | 99.9 |
| MULTIMODAL | CHATGPT 4O | 75.9 |
| | DOUBAO-1.5-VISION-PRO | 76.0 |
| | DEEPSEEK-VL2-SMALL | 5.9 |
| | LLAVA-NEXT-VIDEO-7B-HF | 37.4 |
| | QWEN2-VL-7B-INSTRUCT | 69.4 |

The variance in accuracy highlights the disparity in reasoning or belief assessment capabilities among these models. This indicates that model architecture, training data, or multimodal integration plays a critical role in achieving higher performance in such tasks. The deepseek-vl2-small model achieved only 5.9% accuracy rate on 1944 initial belief tasks, and the reason for this low error rate was that 89.9% were invalid responses.

# E. Hyperparameters' analysis in Section 4.4

The impact of hyperparameters $K$ and $\alpha$ on intervention strength is shown in Figures 15 to 18. We treat generated invalid responses as incorrect answers. Across all intervention results, the intervention direction based on the protagonist's perspective achieves the best performance, which aligns with our expectations and is applied in our experiments.

Specifically, Figures 15 to 18 illustrate a wide span of hyper-parameter settings. We find that the effect of the intervention is confined to a valid interval; once this interval is exceeded, the MLLMs' responses deteriorate. The parameter $\alpha$ remains effective roughly within the range [–50, 50], and the choice of $K$ is informed by the number of hidden heads in the MLLMs. Within the valid region, these two hyper-parameters affect model performance by no more than 10% on average, and their tuning produces a smoothly varying perturbation until the edge of the valid interval is reached. These results show that our approach relies on an interpretable intervention based on internal representations and input perturbations, rather than on hyper-parameter tuning, and that it is not highly sensitive to the specific hyper-parameter values chosen.

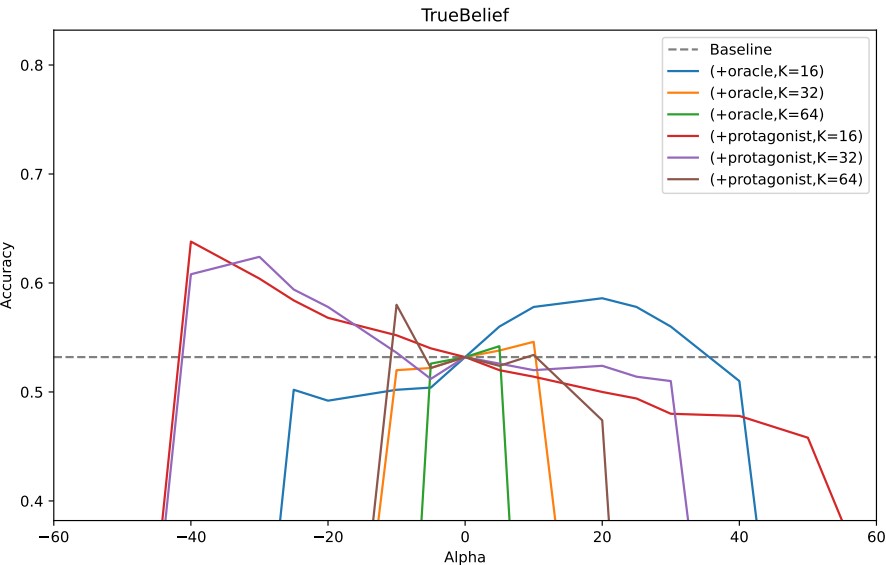

*Figure 15.* The impact of the hyperparameters $K$ and $\alpha$ on the LLaVA-NeXT-Video-7B-hf model on the First-order TB task.

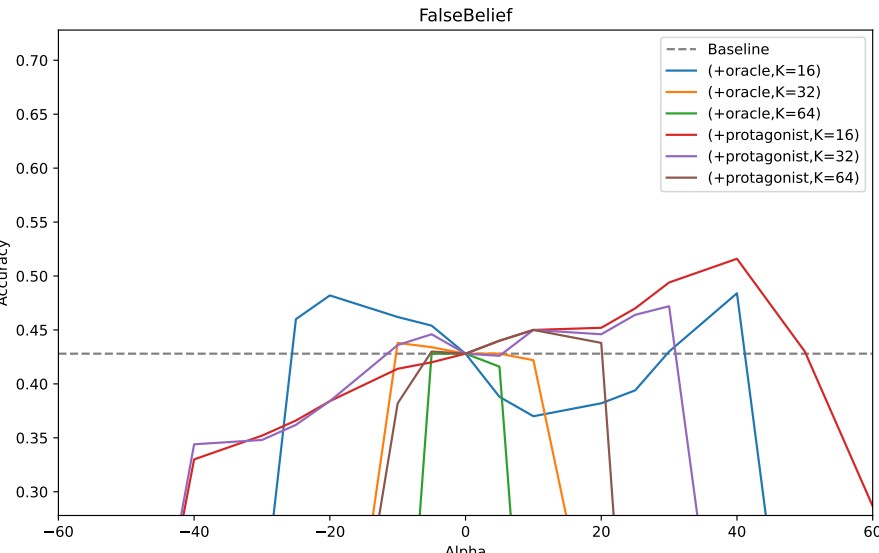

*Figure 16.* The impact of the hyperparameters $K$ and $\alpha$ on the LLaVA-NeXT-Video-7B-hf model on the First-order FB task.

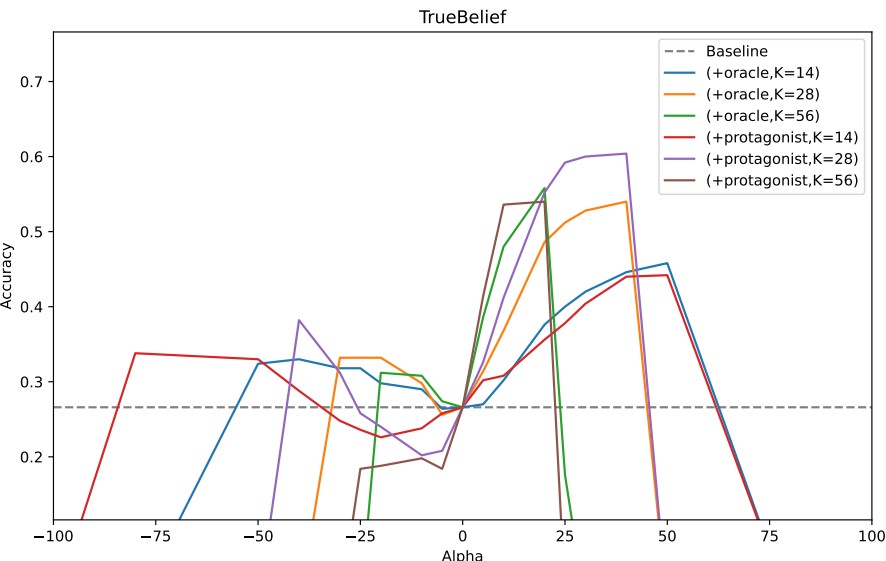

*Figure 17.* The impact of the hyperparameters $K$ and $\alpha$ on the Qwen2-VL-7B-Instruct model on the First-order TB task.

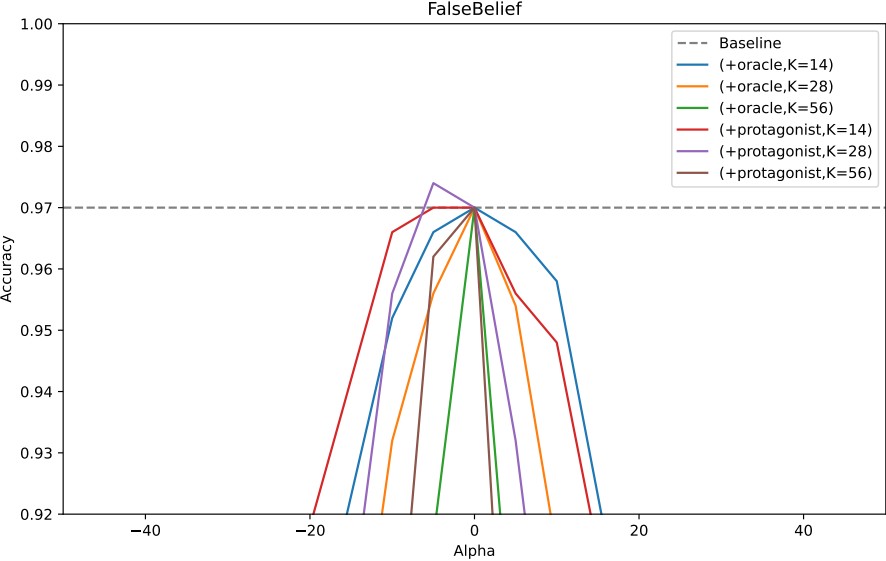

*Figure 18.* The impact of the hyperparameters $K$ and $\alpha$ on the Qwen2-VL-7B-Instruct model on the First-order FB task.

## F. Evaluation protocol of baseline test

Our objective is to provide MLLMs with complete third-person perceptual information in both visual and textual formats (representing an omniscient perspective) and require MLLMs to separate perceptual information corresponding to different perspectives. This allows the models to infer the correct beliefs from each perspective.

Following the standard zero-shot settings for ToM QA evaluations as described in the literature (Shapira et al., 2024), we assess all models without any additional training. The evaluation includes questions related to initial beliefs, first-order beliefs, and second-order beliefs. The evaluation metrics include the accuracy of correctly answering TB, FB, and both TB

and FB simultaneously.

### F.1. Objective

The primary goal of this evaluation is to provide MLLMs with complete third-person perceptual information in both visual and textual formats, representing an omniscient perspective. MLLMs are tasked with separating perceptual information corresponding to different perspectives, enabling them to infer correct beliefs associated with each perspective.

### F.2. Setup

In line with the standard zero-shot settings for ToM QA evaluations, as outlined in the literature (Shapira et al., 2024), all models are assessed without any additional training or fine-tuning. This ensures that the evaluation reflects the inherent ToM reasoning capabilities of the models without being influenced by dataset-specific optimizations.

### F.3. Evaluation Scope

The evaluation employs the following accuracy metrics to measure the model's performance:

**Accuracy of initial belief test** Measures the model's ability to correctly understand the scenario.

**TB Accuracy of first order belief test** Evaluates the model's performance in identifying true beliefs within first-order reasoning scenarios.

**FB Accuracy of first order belief test** Assesses the model's capacity to correctly infer false beliefs in first-order reasoning tasks.

**TB Accuracy of second order belief test** Tests the model's ability to discern true beliefs in second-order reasoning contexts.

**FB Accuracy of second order belief test** Measures the model's effectiveness in identifying false beliefs in second-order reasoning scenarios.

The model's responses are scored based on their ability to correctly answer questions in each belief category. Each category and the performance of both together are reported separately.

## G. Additional Probing Results

We present the full probing results in first-order belief task and second-order belief task for both models using logistic regression models in Figure 19 and Figure 20. The probing accuracies vary across models and tasks.

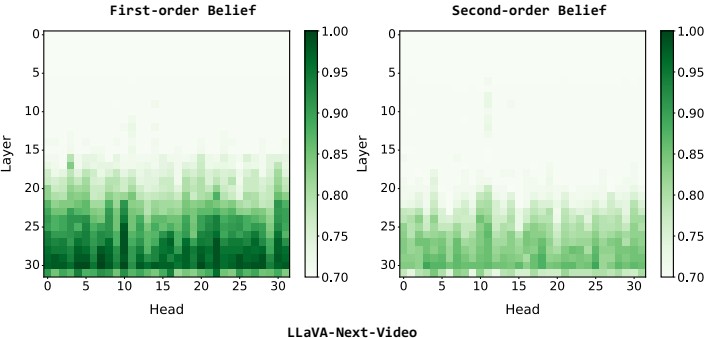

*Figure 19.* Probe accuracies on first-order belief task and second-order belief task based on the attention head activations in all layers of LLaVA-Next-Video.

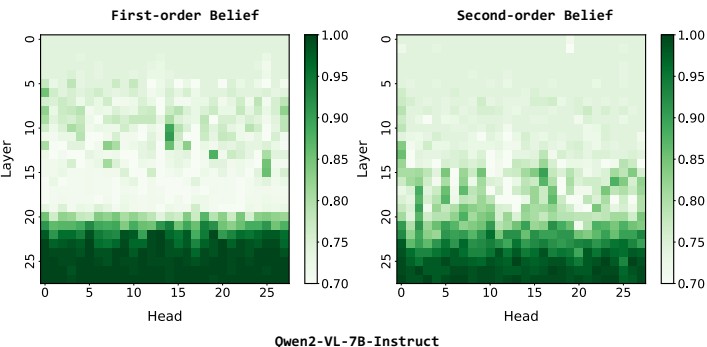

*Figure 20.* Probe accuracies on first-order belief task and second-order belief task based on the attention head activations in all layers of Qwen2-VL-7B-Instruct.

## H. Probing on Different Dataset (MMToM-QA)

We further validated the effectiveness of our method on the MMToM-QA dataset. The MMToM-QA dataset consists of 134 videos, capturing an individual searching for everyday objects in a home environment. This aligns with cognitive science research on mental state attribution in navigational agents.

On average, each video contains 1,462 frames and depicts 36 types of human behaviors. Based on these videos, the dataset includes 600 questions designed to assess both goal reasoning and belief reasoning abilities. Each question is paired with a video clip representing the complete activity (e.g., RGB-D frames), a textual description of the scene, and the actions taken by the individual in the clip. The questions follow a binary-choice format and are categorized into seven reasoning types (as detailed in the original dataset documentation). Specifically, the belief reasoning task consists of 300 questions (100 per type), while the goal reasoning task comprises 300 questions (75 per type). Additionally, the dataset provides 1,000 procedurally generated videos, annotated with ground-truth information on scenes, objects, goals, and beliefs for model training.

In our experiments, we utilized only the belief reasoning subset of the dataset (Figure 21). However, due to the absence of explicit positive-negative video pairs, we manually curated and filtered the dataset, constructing first-order TB and FB samples. This refinement enables a more precise evaluation of the model's ToM reasoning capabilities.

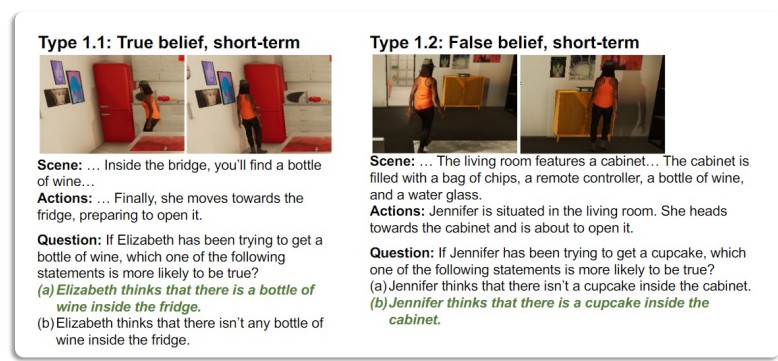

*Figure 21.* Sample examples from the MMToM-QA dataset. The question types utilized in MMToM-QA are also illustrated.

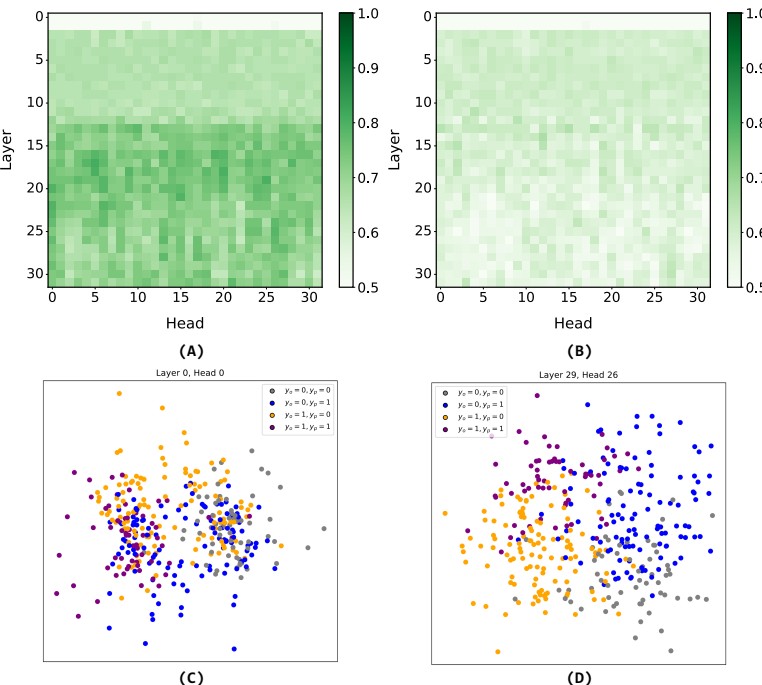

*Figure 22.* (A) Omniscient. (B) Protagonist. The linear probing accuracy of all heads across all layers in LLaVA-Next-Video on the test set. (C) Insensitive. (D) Sensitive. The linear separability of belief representations is explained through a visual interpretation of the typical representation space.

# I. Dataset Construction Pipeline

Our dataset is produced almost entirely through automated generation and verification, with only minimal manual annotation and rigorous quality checks. Although Theory-of-Mind (ToM) reasoning is intrinsically complex, our script-driven workflow guarantees consistent alignment among visual inputs, agent actions, and narrative descriptions.

## I.1. Construction and Annotation

**Map design.**  We manually created 27 distinct $10 \times 7$ maps in Excel, each with 3 rooms and unique layouts.

**Automated validation and rendering.**  Map validity was verified with Python scripts (e.g., enclosed rooms, door placement). Then, using the MultiGrid library, we rendered maps with:

- **Colour palette:** assigned from 6 highly distinguishable colors (red, green, blue, yellow, purple, white).

- **Agent placement:** two groups of agents were randomly placed in hallways with colors distinct from rooms; initial orientations were randomized.

- **Path planning:** agent trajectories were generated using Breadth-first search to ensure valid, logical movement without dead ends.

**Task generation.**  The combination of different variables results in 648 basic samples. For each sample, we generate both "door open" (TB) and "door closed" (FB) conditions, totaling 1296 samples. Second-order belief tasks follow the same structure with minor narrative adjustments.

### I.2. Quality Assurance

- **Automation-first:** key elements (layout, paths, doors, task type) were generated and verified via script, minimizing subjective error.

- **Human review:** we manually reviewed samples for layout issues, trajectory logic, and narrative coherence.

- **Staged execution:** tasks were divided into three stages with controlled timing to ensure logical, coherent event flow.

- **Controlled variables:** we used unified logic for all visual and script elements, systematically varying only key factors (room order, agent orientation, colors, door state).

### I.3. On ToM Difficulty and Dataset Validity

- **Controlled scenarios:** carefully constrained scenes reduce noise, allowing clearer focus on ToM and multimodal reasoning.

- **Scalability:** current difficulty is moderate and sufficient for analyzing belief reasoning. We plan to expand with more complex scenarios in future releases.

