# OpenReview forum: "From Black Boxes to Transparent Minds: Evaluating and Enhancing the Theory of Mind in Multimodal Large Language Models"
_ICML.cc/2025/Conference — ICML 2025 poster_

### Official Review · Reviewer_r21H · 2025-03-10

**Overall Recommendation:** 3

**Summary:**

This paper studies MLLMs’ ability on theory of mind. The authors first construct a benchmark testing MLLMs’ first-order and second-order theory of mind based on grid world setting, and then probe MLLMs’ understanding of beliefs with linear probing. Experiments show that some attention heads show distinguishment of true and false beliefs. Finally, the authors propose improving MLLM theory of mind through attention calibration.

In summary, the authors study a valuable topic, but the proposed benchmarks is flawed. The analysis and proposed method is not novel and efficient enough. A revision of the benchmark, further analysis, and a method with more novelty can improve the paper.

**Claims And Evidence:**

The authors compare the performance of different MLLMs and humans on the proposed benchmarks and report all the results. However, there are some concerns.

How do the authors define “robustness”? does humans’ high accuracy really imply the robustness of the proposed benchmarks?
What is the definition of “error-transfer” nature?

**Essential References Not Discussed:**

The authors do not point out previous grid-based benchmarks (L149-L153), making it hard for readers to understand related work. If the authors are the first to propose grid-based benchmarks, they should claim confidently. If the authors are not, they should explicitly cite previous grid-based benchmarks.

The authors did not discuss works on general MLLM benchmarks such as MMBench [1], SEED-Bench [2], MV-Bench [3], MM-Vet [4], R-Bench [5], etc.

[1] Liu, Yuan, Haodong Duan, Yuanhan Zhang, Bo Li, Songyang Zhang, Wangbo Zhao, Yike Yuan et al. “Mmbench: Is your multi-modal model an all-around player?.” In Computer Vision – ECCV 2024: 18th European Conference, Milan, Italy, September 29–October 4, 2024, Proceedings, Part VI, pp. 216-233. Cham: Springer Nature Switzerland, 2024.

[2] Li, Bohao, Yuying Ge, Yixiao Ge, Guangzhi Wang, Rui Wang, Ruimao Zhang, and Ying Shan. “Seed-bench: Benchmarking multimodal large language models.” In Proceedings of the IEEE/CVF Conference on Computer Vision and Pattern Recognition, pp. 13299-13308. 2024.

[3] Li, Kunchang, Yali Wang, Yinan He, Yizhuo Li, Yi Wang, Yi Liu, Zun Wang et al. “Mvbench: A comprehensive multi-modal video understanding benchmark.” In Proceedings of the IEEE/CVF Conference on Computer Vision and Pattern Recognition, pp. 22195-22206. 2024.

[4] Yu, Weihao, Zhengyuan Yang, Linjie Li, Jianfeng Wang, Kevin Lin, Zicheng Liu, Xinchao Wang, and Lijuan Wang. “MM-Vet: Evaluating Large Multimodal Models for Integrated Capabilities.” In International Conference on Machine Learning, pp. 57730-57754. PMLR, 2024.

[5] Wu, Mingrui, Jiayi Ji, Oucheng Huang, Jiale Li, Yuhang Wu, Xiaoshuai Sun, and Rongrong Ji. “Evaluating and Analyzing Relationship Hallucinations in Large Vision-Language Models.” In International Conference on Machine Learning, pp. 53553-53570. PMLR, 2024.

**Experimental Designs Or Analyses:**

The setting on video-only evaluation is flawed. If the testees are not told that agents can not know what is happening outside when the door is closed, they will make mistakes intrinsically. Such errors do not mean that they do not have good theory of mind. Instead, they are misled by the authors.

In Suppl. Fig. 11 and Fig. 12, the authors give two different captions to one video snapshot, which is insensible. The same video can not represent the situations where there is a timing and where there is no timing. This also shows the intrinsic flaws in the video-only setting.

In Suppl. Fig. 12 and Fig. 13, the authors refer to “green agent”, but there is no green agent in the setting. The “Question” asks about the white agent’s belief, but the “belief true” and “belief false” show the yellow agent’s belief. Therefore, there are typos or maybe even wrong annotations in the presented data example.

In Suppl. Sec. B, the authors do not explain the meaning of 𝑦𝑝𝑟𝑜𝑡𝑎𝑔𝑜𝑛𝑖𝑠𝑡, 𝑦𝑝𝑎𝑟𝑡𝑖𝑐𝑖𝑝𝑎𝑛𝑡, and 𝑦𝑜𝑚𝑛𝑖𝑠𝑖𝑒𝑛𝑡, making it hard for readers to understand the details. The content in the “Belief” row may reflect the authors’ misunderstanding of first-order and second-order belief. The authors should further clarify.

**Methods And Evaluation Criteria:**

The benchmark can test MLLMs’ theory of mind under proper settings.
The authors proposed a method to improve MLLM theory of mind. However, the proposed methods is too straightforward and lacking in novelty. The figure in Suppl. Sec. E shows that the method relies heavily on hyperparameter tuning and can not show improvement on TB and FB tasks at the same time.

**Other Comments Or Suggestions:**

Some cited papers have published versions such as [1, 2]. The authors should cite published versions to prove that their research field is well-recognized by the community.

[1] Shapira, Natalie, Mosh Levy, Seyed Hossein Alavi, Xuhui Zhou, Yejin Choi, Yoav Goldberg, Maarten Sap, and Vered Shwartz. “Clever Hans or Neural Theory of Mind? Stress Testing Social Reasoning in Large Language Models.” In Proceedings of the 18th Conference of the European Chapter of the Association for Computational Linguistics (Volume 1: Long Papers), pp. 2257-2273. 2024.

[2] Sap, Maarten, Ronan Le Bras, Daniel Fried, and Yejin Choi. “Neural Theory-of-Mind? On the Limits of Social Intelligence in Large LMs.” In Proceedings of the 2022 Conference on Empirical Methods in Natural Language Processing, pp. 3762-3780. 2022.

**Other Strengths And Weaknesses:**

See the reviews above.

**Questions For Authors:**

See above.

**Relation To Broader Scientific Literature:**

The key contributions are related to MLLM benchmarking, MLLM representation analysis, and MLLM hallucination mitigation.

**Theoretical Claims:**

There are no theoretical claims in this paper.

---

> ### Author Rebuttal · Authors · 2025-04-01
>
> Thank you for your helpful and constructive reviews. We respond to your concerns and questions below.
>
> # Claims And Evidence
>
> The term "robustness" refers to the internal consistency of our task design. The GridToM dataset is generated via a fully automated pipeline with systematically controlled scenarios.
>
> Our human experiments are intended to confirm that the auto-generated scenarios and reasoning chains are intuitive and interpretable to humans—thus indirectly validating the method's reliability.
>
> To avoid ambiguity, we will revise "robustness" to "consistency" for clarity.
>
> In L331, our intention was to reference the psychological "unexpected transfer task", where false beliefs arise from unobserved changes. The weaker TB performance may be due to the model overgeneralizing from false-belief patterns and ignoring key multimodal cues.
>
> We will revise the terminology to avoid ambiguity.
>
> # Methods And Evaluation Criteria
>
> To our knowledge, we are the first to enhance MLLMs' ToM abilities through targeted attention direction detection and intervention. Our method is not highly sensitive to the choice of hyperparameters. The large range of the hyperparameter Alpha was selected to test the model's limits (interference beyond this range could lead to the failure of all responses).
>
> Our method improves VLMs' ToM performance through interpretable interventions based on internal representations and input perturbations, not hyperparameter tuning. It requires no training or architecture changes, making it generalizable to other reasoning tasks.
>
> Appendix E explores a wide hyperparameter range. We found that the effect is limited to a valid interval—outside of which responses fail. Alpha remains effective within the range of [-50, 50], while the choice of K depends on the number of hidden heads in the model and shows consistent effectiveness across all intervention tasks. Within the valid range, both hyperparameters affect model performance by no more than 10% on average.
>
> In the revision, we will emphasize this stability and avoid confusion about performance fluctuations from extreme values.
>
> We also demonstrate gains on both TB and FB tasks, as shown by the BOTH metric in Table 1.
>
> # Experimental Designs Or Analyses
>
> Testees were not misled. In the video-only condition, core rules—including that closed doors block perception—were clearly explained. The removed text was narrative only and did not affect task understanding. We will clarify this in the revision.
>
> We acknowledge a caption error in Fig. 11. It depicts a first-order belief task for both TB and FB, as correctly stated in the main text (L767–L769). Reviewer ummn also noted this.
>
> Fig. 12 includes two captions for two second-order TB videos, as shown in Fig. 7. All videos include temporal annotations.
>
> In L656, the timing setup refers to the presence of a timeline and corresponding timestamps for events in all scenarios. For example, whether the yellow agent's door is closed at the moment the white agent enters the red room determines whether the white agent can correctly infer the yellow agent's belief.
>
> Fig. 12 and Fig. 13 contain textual errors in the appendix. The second question in Fig. 12 and the first in Fig. 13 should read: "Where does the white character think the yellow character thinks the white character should be?"
>
> The mention of a green character is a typographical error limited to the appendix. The main text and experimental setup are correct. We will fix this in the final version.
>
> Regarding "Question": Fig. 12 and Fig. 13 illustrate second-order belief reasoning, which involves inferring what one agent believes about another agent's belief. The content of the second-order belief is, by definition, the first-order belief itself. This is not an error.
>
> As for Suppl. Sec. B, we acknowledge that some symbols were not explicitly defined. However, their meanings are clearly explained in Section 4.2:
>
> - $y_{protagonist}$: protagonist's belief label
> - $y_{participant}$: participant's belief label
> - $y_{omniscient}$: omniscient's belief label
>
> The "Belief" row shows true/false belief combinations across perspectives, essential for second-order reasoning. These tasks involve nested beliefs (e.g., what the participant believes about the protagonist's belief), forming a natural 2×2 structure, as shown in Fig. 7 and Suppl. Sec. B.2.
>
> # References Not Discussed
>
> Benchmarks like MMBench focus on general MLLM capabilities, while our work targets ToM reasoning—nested beliefs, perspective-taking, and multi-agent cognition—which these benchmarks do not cover. Our task-specific, cognitively grounded framework includes structured annotations for probing reasoning depth. We will mention these benchmarks to highlight the distinction and complementarity of our approach.
>
> # Other Comments
>
> Thank you for pointing out relevant related works. We will update Shapira et al. to its EACL 2024 version and add Sap et al. (EMNLP 2022) to the related work section.

---

> > ### Comment · Reviewer_r21H · 2025-04-02
> >
> > ## Claims and Evidence
> >
> > Thanks for the detailed explanation.
> >
> > ## Methods and Evaluation Criteria
> >
> > Thanks for the detailed explanation. I do notice that in Fig. 17 most settings can help improve Qwen2's performance on TB task. However, I also noticed some other phenomena in the figures. For example, in Fig. 15, "+protagonist,K=16,$\alpha$=-40" achieves best performance on First-order TB task. However, the same setting undermines LLaVA-NeXT-Video-7B-hf's performance on first-order FB task, as shown in Fig. 16. The same for many other settings for LLaVA-Next-Video-7B-hf. Similarly, in Fig. 18, most settings undermine the performance on First-order FB task. How do the authors explain these phenomena? It seems that these phenomena are in conflict with Tab. 1.
> >
> > ## Experimental Designs Or Analyses
> >
> > Thanks for the detailed explanation. Most of my concerns have been addressed. But I am still concerned whether the two videos for second order FB tasks in Fig. 7 are substantially different so that the models and testees will not be misled. Can the videos explicitly show the timing? 4 frames selected from the source videos are inadequate. A figure like Fig. 8 and Fig. 9 will be better.
> >
> > The correspodance between videos and captions is in chaos. Only after the authors' explanation can I understand. I suggest that the authors should correct all the typos and explicitly claim the correspondance between video clips and captions, so that the readers can understand without difficulty.
> >
> >
> > Thanks for the following response. All my concerns has been solved, I increased my score.

---

> > > ### Author Response · Authors · 2025-04-04
> > >
> > > Thank you for your prompt response. In regard to the questions and concerns you raised, we provide the following reply.
> > >
> > > # Methods and Evaluation Criteria
> > >
> > > We believe that the phenomenon you mentioned actually corroborates the conclusions presented in Table 1. Regarding the identical settings in Figures 15 and 16, for instance, "+protagonist, K=16, $\alpha$=-40," the LLAVA-NEXT-VIDEO-7B-HF model exhibits a "symmetric" performance following interference. We interpret this as an indication of the "contradictory nature" between the TB and FB tasks, which is consistent with the results of other models listed in Table 1 (e.g., ChatGPT 4o, DOUBAO-1.5-VISION-PRO, QWEN2-VL). These models face difficulties when simultaneously evaluating both TB and FB tasks. Specifically, these models show high sensitivity to the FB task, and this sensitivity impacts the model’s judgment on the TB task, leading the model to exhibit beliefs that should only appear in the FB task when faced with the TB task. Our benchmark tests intuitively highlight this challenge.
> > >
> > > Furthermore, Figure 18 demonstrates the effect of our method on the interference for the QWEN2-VL model in the FB task. Since this model is more sensitive to the FB task (FB task accuracy: 97.0%, TB task accuracy: 26.6%), although our method provides limited improvement on the FB task, it successfully maintains relatively stable performance within a reasonable range (0.92–0.98), ensuring a limited decrease in accuracy. Meanwhile, QWEN2-VL also exhibits the aforementioned "contradictory nature," even though its performance is below the baseline. The newly scaled figure with the axes can be viewed here: https://anonymous.4open.science/r/icml25-CE4F.
> > >
> > > Based on the above discussion, we attempted to use the same interference settings with the opposite Alpha hyperparameter (For the LLAVA-NEXT-VIDEO-7B-HF model, we used the following settings: TB task: +protagonist, K=16, $\alpha$=-40; FB task: +protagonist, K=16, $\alpha$=40). As a result, we obtained a BOTH metric of 34.4% (with TB accuracy at 63.8% and FB accuracy at 51.6%), and the QWEN2-VL model improved to 55.6%. However, we maintain that the original results (with the same Alpha hyperparameter settings) serve as an objective discussion point, as this intriguing phenomenon emerged from the experiments and lacks theoretical support at this stage. Nonetheless, your question has made us realize that this phenomenon warrants further discussion to advance the community’s understanding, and we will include this in the revised manuscript.
> > >
> > > # Experimental Designs Or Analyses
> > >
> > > Yes, the videos in Figure 7 are all different; in fact, Figure 7 contains four distinct videos. By observing the states of the room switches in the green and red rooms at frames 0, 11, 19, and 36, it is evident that they are all different. A key event influencing the beliefs of both agents is whether the green room door, where the yellow agent is located, is closed when the white agent first enters the red room, and whether the white agent observes the closing of the green room door. This directly determines the type of second-order ToM task.
> > >
> > > In our experiment, we actually used 7 frames (L139-142), including 4 key frames and 3 intermediate frames. The table in Figure 7 is designed to show the state of the room switches for the four key frames to highlight the crucial events, while the 3 intermediate frames maintain the visual story coherence for the MLLMs. We selected 7 frames for two main reasons: first, to ensure that the input token count does not exceed the maximum token limit of any MLLM to avoid automatic truncation of information; second, to keep the input information both concise and complete. Through experiments on Initial Belief, First-order Belief, and Second-order Belief (Tables 3 and 1), we verified that MLLMs can capture complete visual information with the 7 frames.
> > >
> > > Additionally, we will correct all typographical errors and provide clearer explanations of the dataset.
> > >
> > > **Once again, thank you for all your valuable suggestions.**
> > >
> > > Thank you for the updated score and for taking the time to consider our rebuttal. We sincerely appreciate your recognition. We will make sure to address all the issues you previously raised in the final version of the paper.

---

### Official Review · Reviewer_NkZp · 2025-03-11

**Overall Recommendation:** 4

**Summary:**

This paper develops a new approach to evaluate Theory of Mind (ToM) abilities of Large Language Models. Taking as a starting point the potential limitations of previous ToM experiments (difficulties in capturing an agent’s perception, ToM tasks not addressing internal representations), it designs a specific test environment (GridToM), in which context and cognitive perspective information can be controlled. This benchmark is used to test five state-of-the-art multimodal LLM with human raters as a baseline. Subsequently, the authors apply an activation inference strategy to two of the LLMs (LLAVA-NEXT-VIDEO-7B-HF, QWEN2-VL-7B-INSTRUCT) resulting on improved ToM performance across tasks.

**Claims And Evidence:**

The main objective is benchmark development, for which claims are validated through LLM performance comparison and human users ground truth.

**Essential References Not Discussed:**

Verma, M., Bhambri, S. and Kambhampati, S., 2024, March. Theory of mind abilities of large language models in human-robot interaction: An illusion?. In Companion of the 2024 ACM/IEEE International Conference on Human-Robot Interaction (pp. 36-45).

(secondary):
Mao, Y., Liu, S., Ni, Q., Lin, X. and He, L., 2024. A review on machine theory of mind. IEEE Transactions on Computational Social Systems, vol 11, n.6, Dec 2024.

**Experimental Designs Or Analyses:**

GridToM implements the unexpected transfer task in an ‘ARC’ benchmark esthetic. This design supports a context in which to articulate different agent’s viewpoints (perspective separation) and belief inference.
Making second-order (meta-cognitive) aspects accessible to the benchmark is definitely advancing the state-of-the-art of LLM ToM investigation.
The use of attention heads to distinguish information across perspectives is intellectually compelling, and provides demonstrable enhancements. It may be worthy of its own investigation but I would still suggest to keep the early results, even on a subset of LLMs, in the final version of the paper.

**Methods And Evaluation Criteria:**

The paper is itself on an evaluation method and in my view follows best practice, in particular in view of context generation through the 'dynamics' of the environment, which produced multiple instances of ToM problems, from a consistent set of principles.
Results are clearly and transparently reported for both first-order and second-order aspects.
There is a good sample of LLM being tested across parameter size and instruction-tuned types of models.
One limitation could be that the binary options in the benchmark may contribute in part to the high performance observed.

**Other Comments Or Suggestions:**

The paper shows an appropriate balance between the core discussion and supplementary material, which is useful considering the experimental design and its multimodal content. It is generally well illustrated considering the sophistication of the experiments.
Still on presentation, I was less convinced about the TARS example; I recognize the pedagogic value, although the ToM benchmark issue is complex enough to deserve additional explanations rather than analogies. Whether it should be modified is however left to the authors’ discretion.
Post-rebuttal: the authors have answered my questions and responded satisfactorily; I remain positive about this paper.

**Other Strengths And Weaknesses:**

I would not identify major weaknesses in the paper. When limitations are considered, it could be appropriate to relate this approach to the work of Verma et al. [2024] in particular on the perceived behavior recognition issue: to which extent the benchmark addresses a subset of it or might scale up to more complex behaviors.

**Questions For Authors:**

Is the attention-based enhancement dependent somehow on modalities?
How could you extend the narrative scenarios beyond binary choices for beliefs?

**Relation To Broader Scientific Literature:**

ToM abilities for LLM is a debated topic since the 'Sparks of AGI' paper [Bubeck et al., 2023]. This paper is relevant to most papers to date in offering alternatives to Sally-Anne type testing and experiments based on tests such as [Strachan et al., 2024]. It is less connected to ToM robotics papers, though.

**Theoretical Claims:**

N/A

---

> ### Author Rebuttal · Authors · 2025-04-01
>
> We sincerely appreciate your recognition of our work and thank you for the thoughtful suggestions. We respond to your concerns and questions below.
>
> # Methods And Evaluation Criteria
>
> Our benchmark is designed as a foundational framework, using binary labels to establish clear distinctions between correct and incorrect beliefs. Introducing a wider range of options and more comprehensive evaluation schemes is indeed a valuable direction. We believe that the evaluation of multiple options could potentially be extended using a binary classification approach, where multiple negative samples are categorized as generalized erroneous beliefs. This will be part of our future research plans.
>
> # Experimental Designs
>
> Thank you very much for recognizing the value of using attention heads to distinguish different perspectives and enable second-order belief reasoning. We agree that this approach is highly worthy of further investigation and have planned to conduct broader model validation and more detailed analysis in our future work. At the same time, we will adopt your suggestion to retain the early results and include these preliminary experimental conclusions in the final version of the paper.
>
> # Relation To Broader Scientific Literature
>
> We understand your point about the current paper’s insufficient connection to embodied intelligence and the field of robotics. In fact, we are also deeply interested in embodied intelligence. This work represents our initial effort to explore the Theory of Mind (ToM) capabilities of MLLMs. We plan to extend this approach to more interactive and visual scenarios in future work, aiming to build a closer bridge between ToM in MLLMs and embodied intelligence in robotics.
>
> # Essential References Not Discussed & Other Strengths And Weaknesses
>
> Thank you for pointing out relevant related works. Although we have not discussed Verma et al. (2024) and Mao et al. (2024) in the current version, we recognize their relevance and value. We will include discussion of these papers in the revised manuscript to better position our work within the broader context of ToM research in LLMs, particularly as it relates to human-machine interaction and embodied intelligence.
>
> # Other Comments Or Suggestions
> Thank you for the feedback. Our original intention was to use a vivid example to illustrate higher-order beliefs. We acknowledge that ToM benchmarks involve complexities beyond simple analogies. Following your suggestion, we will add a more detailed explanation of ToM in the revision.
>
> # Questions for Authors
> **1. Modality Independence:**
>
> We believe that attention-based enhancement is not strictly dependent on a specific modality but rather on carefully designed experiments. Prior work and our own results show the effectiveness of attention enhancement across both text-only and multimodal settings. Our experiments (Sup. G and H) demonstrate generalization across GridToM and MMTOM, suggesting the potential for extending this method to other modalities such as audio. Additionally, We believe attention-based enhancement depends on carefully designed experiments to reduce the impact of noise in attention signals.
>
> **2. Binary Testing in ToM Research:**
>
> ToM studies often rely on structured binary choices (e.g., true/false) to assess belief reasoning, as seen in classic tasks like the “unexpected location” and “unexpected contents” paradigms. Existing benchmarks for LLMs, such as ADV-CSFB (text-based) and MMTOM-QA (multimodal), also follow this approach. Our study continues this tradition. That said, exploring more open-ended or multi-choice formats could test reasoning flexibility in more complex settings, though this would introduce higher demands in task design and annotation. We see this as a promising direction for future research.
>
> Once again, thank you for your kind support and constructive feedback.

---

> > ### Comment · Reviewer_NkZp · 2025-04-04
> >
> > Thank you for provided detailed follow-up comments, which confirm my positive appreciation of the paper.
> > You answered both questions to my satisfaction, and the reference to other ToM benchmarks has allayed any remaining concerns on binary testing.

---

> > > ### Author Response · Authors · 2025-04-07
> > >
> > > Thank you for your positive evaluation and thoughtful review of our work. We truly appreciate your constructive comments and will make sure to address all the points you raised in the final version of the paper.

---

### Official Review · Reviewer_ummn · 2025-03-15

**Overall Recommendation:** 3

**Summary:**

This paper aims to explore the Theory of Mind (ToM) capabilities for multimodal large language models (MLLMs).
To this end, it proposes a new dataset, GridToM, which is designed to evaluate MLLM Tom reasoning from multiple perspectives.
Based on GridToM, they then conduct experiments using different tech to detect the ToM in MLLMs in a zero-shot setting.
Experimental results and analyses show the attention heads in MLLMs can be capable of distinguishing such mental states.

**Claims And Evidence:**

Most claims are well supported.

**Essential References Not Discussed:**

NA

**Experimental Designs Or Analyses:**

The paper only experiments with the zero-shot setting, which makes their findings and conclusions less generalizable.
Additionally, as the authors suggested in Sec 5.2, the selection of models is indeed somewhat limited.

Also, from a perspective of dataset construction, there is no validation of the data quality.

**Methods And Evaluation Criteria:**

The major contribution of this paper is the GridToM dataset, which provides manipulable multimodal visual-linguistic causal stories. However, I fail to find any relevant information about how this dataset is built, e.g, do the authors collect data from existing datasets with additional annotation? How do the authors conduct the annotation process? What is the annotators' background? What is the quality control process? Such information is very important to a dataset/benchmark paper, which is not provided in the main text nor the Appendix. Also, ToM is a difficult task even for humans - it makes me doubtful about data quality.

**Other Comments Or Suggestions:**

The figure 3 is difficult to understand without the detailed description of the Appendix, while I believe a good figure should be easy to follow with just the caption information.
Also, for 1st order: why the labels for both TB and FB of omniscient are "Purple" ( which seems contradictory to Figure 11)? and why "Red" for both TB and FB of protagonist?

**Other Strengths And Weaknesses:**

Please see above

**Questions For Authors:**

Please see above

**Relation To Broader Scientific Literature:**

This paper is very relevant to ToM research, and can be important to MLLM reasoning.

**Theoretical Claims:**

No theoretical claims.

---

> ### Author Rebuttal · Authors · 2025-04-01
>
> Thank you for your helpful and constructive reviews. We respond to your concerns and questions below.
>
> # Methods And Evaluation Criteria
>
> We agree that the construction of the GridToM dataset should be described in greater detail. We provide further clarification here and will include the full pipeline both in the main text and the appendix of the revised version. Our dataset is built primarily through automated generation and verification, with minimal manual annotation and thoughtful quality control. Despite the intrinsic complexity of ToM reasoning, our structured, script-driven approach ensures consistency and reliability across visual input, actions, and narratives.
>
> 1. Dataset Construction and Annotation
>
> - Map Design: We manually created 27 distinct 10×7 maps in Excel, each with 3 rooms and unique layouts.
>
> - Automated Checking and Rendering: map validity was verified with Python scripts (e.g., enclosed rooms, door placement). Then, using the MultiGrid library, we rendered maps with:
>
>   - Colors: Assigned from 6 highly distinguishable colors (red, green, blue, yellow, purple, white).
>
>   - Agent Placement: Two groups of agents were randomly placed in hallways with colors distinct from rooms; initial orientations were randomized.
>
>   - Path Planning: Agent trajectories were generated using Breadth-first search to ensure valid, logical movement without dead ends.
>
> - Task Generation: The combination of different variables results in 648 basic samples. For each sample, we generate both “door open” (TB) and “door closed” (FB) conditions, totaling 1296 samples (see L122–127). Second-order belief tasks follow the same structure with minor narrative adjustments.
>
> 2. Annotation and Data Quality
>
> - Automation First: Key elements (layout, paths, doors, task type) were generated and verified via script, minimizing subjective error.
>
> - Human Review: We manually reviewed samples for layout issues, trajectory logic, and narrative coherence.
>
> - Staged Execution: Tasks were divided into three stages with controlled timing to ensure logical, coherent event flow.
>
> - Controlled Variables: We used unified logic for all visual and script elements, systematically varying only key factors (room order, agent orientation, colors, door state).
>
> 3. On ToM Difficulty and Dataset Validity
>
> - Controlled Scenarios: Carefully constrained scenes reduce noise, allowing clearer focus on ToM and multimodal reasoning.
>
> - Scalability: Current difficulty is moderate and sufficient for analyzing belief reasoning. We plan to expand with more complex scenarios in future releases.
>
> # Experimental Designs Or Analyses
>
> We focus on zero-shot evaluation because Theory of Mind research emphasizes testing a model’s ability to reason about others’ mental states without task-specific demonstrations. Zero-shot settings help assess whether such capabilities naturally emerge during pretraining, without overfitting to specific prompts or memorized patterns. In other words, we believe that the abilities revealed by few-shot learning pertain more to the model’s other capabilities rather than ToM abilities themselves. On the other hand, zero-shot remains a widely adopted and informative setting for assessing underlying reasoning abilities in this domain. This approach follows precedent in prior work such as [Shapira et al., 2023; Jin et al., 2024], all of which adopt zero-shot setups to evaluate ToM reasoning in LLMs or MLLMs.
>
> As for model selection, due to the limited availability of MLLMs supporting multi-image inputs at the time, we selected two strong-performing models: LLaVA-Next-Video and Qwen2-VL. We are actively working on extending our experiments to more models and tasks.
>
> # Other Comments Or Suggestions
>
> In the first-order tasks shown in Fig. 3, the "Omniscient" label represents the objective ground truth. Since the omniscient perspective observes the entire event sequence, the belief remains "Purple" for both TB and FB conditions.
>
> Second-order belief reflects whether one agent correctly estimates another’s belief. Due to space limitations, Fig. 3 only presents the case where the first-order belief is false, which is why both TB and FB under the protagonist’s view are labeled as "Red." The complete set of cases, including those where the protagonist’s belief is labeled as "Purple," is detailed in Fig. 7.
>
> We acknowledge that there is a mistake in the caption of Fig. 11. Fig. 11 illustrates a first-order belief task to both TB and FB conditions, as correctly explained in the main text (L767-769). Reviewer r21H also pointed this out. We will address this issue in the revised manuscript.

---

### Decision · Program_Chairs · 2025-05-01

**Decision:**

Accept (poster)

**Comment:**

This paper primarily contributes an evaluation of Theory-of-Mind (ToM) in Vision Language Models. A further minor contribution is a strategy for improving Theory of Mind in VLMs by calibrating the attention heads. This paper received mostly positive reviews. Reviewers find the proposed benchmark well executed (both in terms of how it is constructed and the models evaluated) and follow-up analyses interesting. Reviewers were initially concerned with the quality of the writing, and missing details about the generation pipeline (which is especially important when proposing a new benchmark). There was also some uncertainty about how this work compares to relevant related work. The author response clarified most of these issues (as far as this is possible without submitting a new draft) and with the promised changes the work appears to be in good shape for publication as is recognized by the reviewers.